# Seasonal variation and release of soluble reactive phosphorus in an agricultural upland headwater in central Germany

Michael Rode[1,2], Jörg Tittel[3], Frido Reinstorf[4], Michael Schubert[5], Kay Knöller[6], Benjamin Gilfedder[7], Florian Merensky-Pöhlein[4], Andreas Musolff[8]

[1]Department Aquatic Ecosystem Analysis, UFZ - Helmholtz-Centre for Environmental Research, Brückstr. 3a, 39114 Magdeburg, Germany
[2]Institute of Environmental Science and Geography, University of Potsdam, Potsdam-Golm, Germany
[3]Department of Lake Research, UFZ - Helmholtz-Centre for Environmental Research Brückstr. 3a, 39114 Magdeburg, Germany (ORCID: 0000-0003-1151-1909)
[4]Department for Water, Environment, Construction and Safety, Magdeburg-Stendal University of Applied Sciences, Breitscheidstr. 2, 39114 Magdeburg, Germany
[5]Department Catchment Hydrology, UFZ - Helmholtz Centre for Environmental Research, Permoserstr. 15, 04318 Leipzig, Germany
[6]Department Catchment Hydrology UFZ - Helmholtz Centre for Environmental Research, Theodor-Lieser-Str. 4, 06120 Halle, Germany
[7]University of Bayreuth, Universitätsstraße 30, 95447 Bayreuth, Germany
[8] Department of Hydrogeology, UFZ - Helmholtz-Centre for Environmental Research GmbH, Leipzig, Germany

*Correspondence to:* Michael Rode (michael.rode@ufz.de)

**Abstract.** Soluble reactive phosphorus concentrations (SRP) in agricultural headwaters can display pronounced seasonal variability at low flow, often with the highest concentrations occurring in summer. These SRP concentrations often exceed eutrophication levels but their main sources, spatial distribution, and temporal dynamics are often unknown. The purpose of this study is therefore to differentiate between potential SRP losses and releases from soil drainage, anoxic riparian wetlands and stream sediments in an agricultural headwater catchment. To identify the dominant SRP sources we carried out three longitudinal stream sampling campaigns for SRP concentrations and fluxes. We used salt dilution tests and natural $^{222}$Rn to determine water fluxes in different sections of the stream, and sampled for SRP, Fe and $^{14}$C-DOC to examine possible redox-mediated mobilization from riparian wetlands and stream sediments. The results indicate that a single short section in the upper headwater reach was responsible for most of the SRP fluxes to the stream. Analysis of samples taken under summer low flow conditions revealed that the stream water SRP concentrations, fraction of SRP within total dissolved P (TDP) and dissolved organic carbon (DOC) radiocarbon ages matched those in the groundwater entering the gaining section. Pore water from the stream sediment showed evidence of reductive mobilization of SRP but the exchange fluxes were probably too small to contribute substantially to SRP stream concentrations. We also found no evidence that shallow flow paths from riparian wetlands contributed to the observed SRP loads in the stream. Combined, results of this campaign and previous monitoring suggests that groundwater is the main long-term contributor of SRP at low flow and agricultural phosphorus is largely buffered in the soil zone. We argue that the seasonal variation of SRP concentrations was mainly caused by variations in the proportion

of groundwater present in the streamflow, which was highest during summer low flow periods. Accurate knowledge of the various input pathways is important for choosing effective management measures in a given catchment, as it is also possible that observations of seasonal SRP dilution patterns stem from increased mobilization in riparian zones or from point sources.

## 1 Introduction

Land-to-water diffuse phosphorus emissions caused by intensive agricultural land use are a major cause of eutrophication in streams, rivers and lakes (Bol et al. 2018). Phosphorus losses from headwater catchments are the result of integrated hydrological and biogeochemical processes occurring within the drainage area and in the stream (Bormann and Likens 1967, Bernal et al. 2014). Such headwater P transport processes can exhibit high spatio-temporal variability and are controlled by landscape properties. This high variability is especially well-described for particulate P losses (Bechmann et al. 2008, Bol et al. 2018). However, recent studies suggest that dissolved and colloidal P mobilization from agricultural land can also result in a high seasonal variation in soluble reactive phosphorus (SRP) concentrations in headwater streams (Bol et al. 2018). This variability in seasonal P concentrations is possibly of greater importance than previously assumed (Dupas et al. 2018) and can be related to a change in the proportion of different P sources throughout the year. SRP mobilisation can occur from various headwater compartments, comprising a) preferential flow in soils and tile drainage, b) riparian wetlands in connection with anoxic conditions and a reductive dissolution of Fe (oxy)hydroxides (Tittel et al. 2022), c) stream sediments via the same reductive process or desorption (Kleinman et al. 2003, Gu et al. 2017, Smolders et al. 2017), and d) groundwater systems (Brookfield et al. 2021).

The leaching of dissolved P from soils has been linked to surface-soil P desorption (McDowell and Sharpley 2001) and the application of fertilizer (Chardon et al. 2007). Subsurface transport is often dominated by preferential flow through soil macropores (Simard et al., 2000). In addition, factors such as high soil P sorption saturation ($P_{sat}$) and oxidation-reduction cycles can greatly increase P mobility through soils (Behrendt and Boekhold, 1993; Heckrath et al., 1995). The lateral transport of P is mostly induced by artificial drainage that provides a lateral short cut between subsurface macropores and surface water (Dils and Heathwaite, 1999), but also occurs in association with particular soil characteristics, e.g. sandy soils (Kleinman et al. 2009).

The hydrological variability of riparian wetlands has also been widely shown to influence SRP mobilization via redox conditions in soils. The high water table and low flow velocities that are typical for riparian wetlands during the wet season can create anoxic conditions. This can lead to the reductive dissolution of Fe (oxy)hydroxides (Jeanneau et al., 2014; Knorr, 2013) and hence to solubilization of the P previously adsorbed or co-precipitated onto/within these mineral phases (Zak and Gelbrecht, 2007). Similar findings have been recorded for upland headwater catchments, with increased SRP mobilization via redox processes during periods when groundwater levels are high and intersect organic-rich soils (Dupas et al. 2017a). It is assumed that these temperature dependent biogeochemical processes could lead to P release into streams and rivers during the summer period when $NO_3^-$ is denitrified and thus missing as redox buffer for the reductive mobilization of Fe oxides (Musolff et al. 2017, Dupas et al. 2018).

Stream sediments have potential to remove or release P to the stream water during summer low flow conditions (Simpson et al. 2021). When streams are mainly fed by groundwater, P reactions in the groundwater-stream interface (i.e. hyporheic zone) may control the release of P. Data suggest that anoxic conditions can cause the release of sediment P in streams (Simpson et al. 2021). The highest seasonal concentrations of SRP are often found in summer and during low flow, and are higher in lowland than in upland rivers (Bowes et al. 2003). SRP redox-mediated release from river sediments has been identified in lowland rivers during summer anoxia (Smolders et al., 2017). This mobilization of P from sediments to the water column was found to be related to the molar P/Fe ratio in stream sediments. The authors suggested that the temporal and spatial variability of soluble P in the water body of lowland rivers was mainly related to internal loading, i.e., to the legacy P in the sediment and not to the corresponding variability in emission and dilution rates (Smolders et al. 2017).

Finally groundwater systems can be important sources for the mobilization and transport of P to surface waters. The P concentration in groundwater depends on the geology and long-term land use pattern in the catchment (Brookfield et al. 2020). Long-term agricultural activities can result in soil P concentrations exceeding the soil sorption capacity, which can be released into the groundwater system in dissolved form (Brookfield et al. 2020, Haygarth et al. 2014). In addition to this accumulation of P in groundwater through anthropogenic activities, the extent of water-rock interaction can also impact the P concentration in groundwater. An important factor is the contact time of groundwater with the aquifer matrix, and slower groundwater flow can lead to more interactions between water, rock and microbes. For example, chemical weathering can affect the sorption of P onto aquifer minerals, where the sorption ability of P can decrease with increasing pH (Cornell and Schwertmann 2003).

A recent comparative study in forest and agricultural headwaters revealed that the highest levels of seasonal SRP occurred in headwater streams featuring riparian wetlands or high groundwater levels in the near stream zone (Dupas et al. 2017b). One of the catchments evaluated in Dupas et al. ( 2017b) is the agricultural Schäfertal catchment which is a typical headwater of the central German lower-mountain hard-rock area. Until now it has been unclear which P sources and transfer pathways are responsible for the distinct seasonal pattern of winter lows and summer highs in SRP level, and evidence is lacking on the dominant P transfer pathways. Furthermore, it is not clear how land use (e.g. P status) might impact baseline SRP concentrations, nor which factors control SRP mobilization (e.g. groundwater heads, land use, temperature, redox processes, P status of riparian zones etc.). Analysis of these controlling factors would enable identification of the sources of SRP in a characteristic hard-rock agricultural catchment in a lower mountain range.

The objectives of the present study were: a) to identify the main pathways for SRP transfer into the stream water of an agricultural headwater catchment during low flow conditions (e.g. transfer from deep groundwater, redox-controlled delivery from riparian wetlands, release from stream sediment); b) to localise the major source areas of SRP within the catchment; and c) to explain the mechanisms leading to the development of characteristic seasonal SRP concentrations.

This study aims to spatially localise and quantify gaining and losing water fluxes along the Schäfertal stream using salt tracer injections and longitudinal measurements of stream $^{222}$Rn activities. By combining these spatially distributed water fluxes with longitudinal water-quality measurements, it was possible to quantify proximal SRP fluxes along the whole stream length. To identify the SRP release processes, we conducted additional radiocarbon, DOC and Fe measurements in gaining groundwater

and in the streambed sediments for comparison with the stream water signature. Combined measurement campaigns during summer and winter low-flow conditions enabled us to evaluate the seasonal influence on the behaviour of SRP transport in the study headwater.

## 2 Material and methods

### 2.1 Study site

The Schäfertal stream is located in an agricultural headwater catchment (1.44 km$^2$, Figure 1), in the lower Harz Mountains in central Germany. Elevation ranges from 391 at the outlet to 474 m. The north- and south-facing hillslopes, with an average slope of 11°, are intensively cultivated (crop rotation: winter wheat, triticale, rapeseed, winter wheat); mineral-fertilizer application levels were between 60 kg (rapeseed) and 148 kg (winter wheat) N ha$^{-1}$ y$^{-1}$, and between 11 and 14 kg P ha$^{-1}$ y$^{-1}$ (LLFG 2021, Kistner et al. 2013). In 2018 the cropping system shifted to organic farming and no more mineral fertilizer was

applied. The mid-catchment valley bottom is dominated by grassland with drainage channels; the upstream hilltop is occupied by sparse forest (Figure 1).

The underlying Paleozoic greywacke and Devonian shale are covered by periglacial layers with varying fractions of loess and rock fragments, resulting in complex geomorphological structures through the soil profile (Kistner et al., 2013). The hillslope soils (Altermann and Mautschke, 1970; Graeff et al., 2009) exhibit relatively high porosity and hydraulic conductivity (mean

value 9.95 * 10$^{-6}$ m s$^{-1}$) in the top soil layer, and lower hydraulic conductivity (mean value 2.31 * 10$^{-7}$ m s$^{-1}$) in the base layer at depths below 0.4 m (Graeff et al., 2009). Soil properties ascertained by field soil core sampling are generally homogenous (Schrön et al., 2017), with a certain degree of spatial variability caused by detailed topographic characteristics such as slope position, valley bottom and exposition (e.g. as reported in Ollesch et al., 2005; Anis & Rode, 2015). Additionally, there is an extensive network of artificial tile drains throughout the central valley bottom (see Yang et al. 2021). Fluvisols and Gleysols

dominate in the valley bottom and are partly drained by the tile drainage network. The arable hillslope soils are Gleyic Cambisols and Luvisols, and a small area of forest soils made up of Rankers and Cambisols is found at the top of the catchment (Figure 1) (Yang et al. 2021). It can be assumed that aquifer thickness ranges from 2 m at the top of the hillslopes to 5 m at the valley bottom (2.4 m on average), based on the dominance of Gleysols and Luvisols toward the valley bottom (Yang et al. 2018). The stream itself has a length of 1747 m and a slope of 2 %, with a mean catchment area normalised discharge of 0.33

mm d$^{-1}$ (~5.5 L s$^{-1}$). The stream has a mean width of 0.4 m and a depth of 0.05 m. The substrate consists of fine and mid-granular sand. It is an open canopy stream without riparian trees and high light availability. The forest area of 3 % is restricted to the upper part of the catchment.

Previous research has shown mean soil total phosphorus (TP) content in the top soil layer (3-5 cm) to be ~916 mg P kg$^{-1}$ while water-soluble phosphorus (WSP) content is 13.1 mg P kg$^{-1}$, indicating a strong influence from fertilizer use. The degree of

phosphorus saturation (DPS) is 31.7% (Kistner et al. 2013). It can be assumed that the total phosphorus (TP) concentration changes only slightly over time (Little et al. 2007) while modelling has shown that temporal variation in WSP concentrations

can be high, caused by fertilizer application and crop uptake (Kistner et al. 2013). Calculated short-term declines of DPS in the top soil layer (upper 2cm of soil) can be explained by rainfall events. This suggests a transport of soluble P compounds to deeper soil layers (Kistner et al. 2013). Top-soil WSP concentrations also display high spatial variability, from 2.3 to 37.6 mg

P kg$^{-1}$ (Kistner et al. 2013). The recorded means for soil organic carbon content and pH in the top soil layer are 21.3 g kg$^{-1}$ and 6.39 respectively (Kistner et al. 2013). DOC concentration in soil pore water ranges between 1.2 and 62.6 mg L$^{-1}$ (Ackermann, 2016).

Due to its location in the eastern lee of the Upper Harz Mountains, the catchment sits in a rain shadow of the Brocken Mountain and therefore has a relatively low average annual precipitation of about 629 mm a$^{-1}$ (1991-2020). Precipitation is relatively

evenly distributed over the year, with slightly higher precipitation values in summer. The discharge regime of the Schäfertal stream, however, is dominated by higher discharges in winter, mainly due to snowmelt, and low flow periods in summer (Figure 2). The influence of earlier mining activities further down the catchment on the flow regime and groundwater levels ceased with the mining activities in the beginning of the 1990s, and pre-mining hydrological conditions have returned.


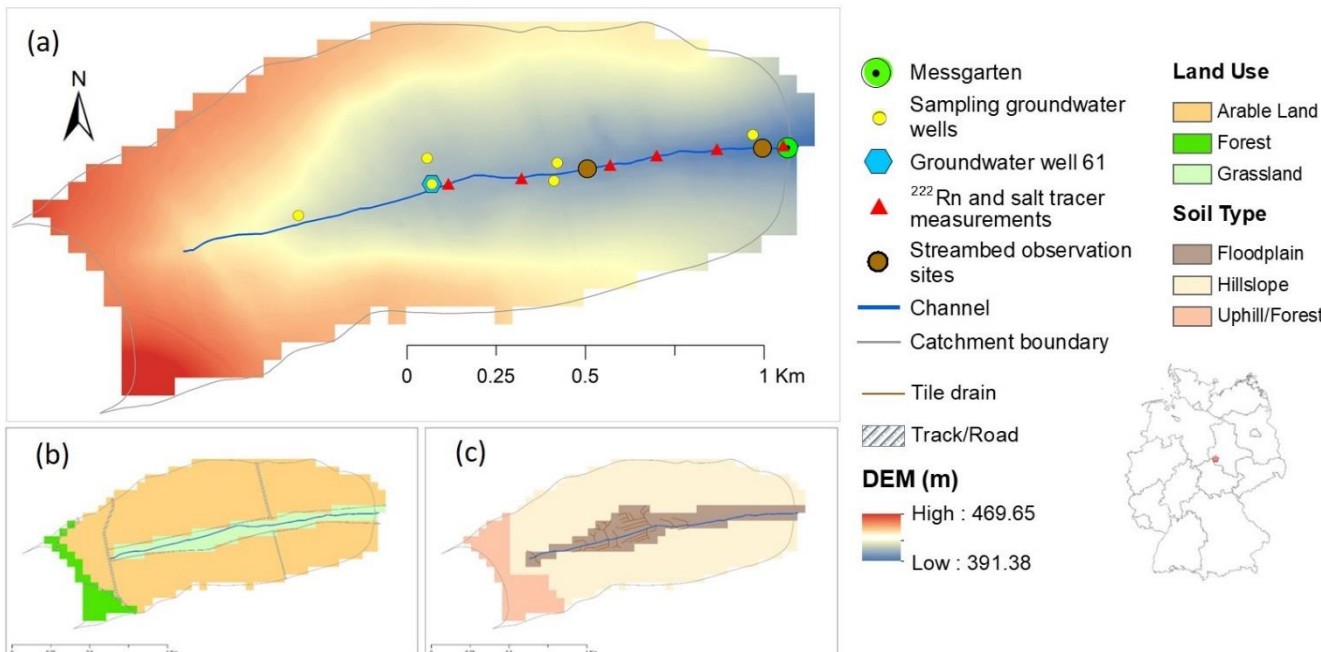

**Figure 1.** The Schäfertal catchment and monitoring locations used by Yang et al. (2021), showing (a) digital elevation model (DEM) and monitoring sites with sampled groundwater wells, (b) land use types, and (c) soil types and the tile drain network. A V-notch weir is installed in the "Messgarten" climate station for discharge measuring and stream water sampling (adapted

from Yang et al., 2021)

In summer, the stream regularly dries up in the area near the source, while it is perennial in the lower sections. The dynamics of groundwater levels near the wetland in the central part of the catchment show a significant decrease due to the relative strong drought conditions during 2018-2019 (well 61, see Figure 1, period 2010-2021, Supplement). An isotope-tracer aided modelling study has shown that under these drought conditions modelled stream runoff from deeper, older storages increased significantly after a particularly wet season, resulting in a sharp increase in mean stream water age (Yang et al. 2021). Earlier long-term stream water quality measurements at the catchment outlet (1999-2010) reveal $NO_3^-$ concentrations in discharge to be between 0.11 and 11 mg N $L^{-1}$ (mean = 4.37 mg $L^{-1}$), DOC concentration between 1.7 and 12.6 mg $L^{-1}$ (mean = 4.23 mg$L^{-1}$), SRP between 0.002 and 0.16 mg $L^{-1}$ (mean = 0.025 mg $L^{-1}$), and TP concentrations between 0.009 and 0.33 mg $L^{-1}$ (mean = 0.067 mg $L^{-1}$). Baseflow stream-concentration data show clear seasonal variations, with $NO_3^-$ peaking in winter while DOC and SRP peak in summer (Dupas et al. 2017b).

## 2.2 Measurement campaigns

Measurement campaigns took place after snowmelt in January 2019 during a period of slightly elevated discharge and groundwater levels, and in September 2019 and 2020 during prolonged periods of low flow and low groundwater levels. These campaigns comprised in-stream salt tracer dilution tests and $^{222}Rn$ measurements in order to analyse lateral inflows to the stream, and water quality measurements to characterize stream water, riparian groundwater and stream sediment properties. Meteorological conditions for all campaigns were characterized by comparatively low rainfall in the preceding days (Figure 2). The groundwater levels in the sampling period ranged between 0.5 and 1.1 m below soil surface. During the three sampling campaigns, mean groundwater levels were 0.65 m (Jan. 2019), 1.0 m (Sep. 2019) and 0.85 m (Sep. 2020) below the surface (see Figure S, Supplement). Note that screens of the wells and thus groundwater sampling depths are between 3.25 m and 10.88 m below soil surface.

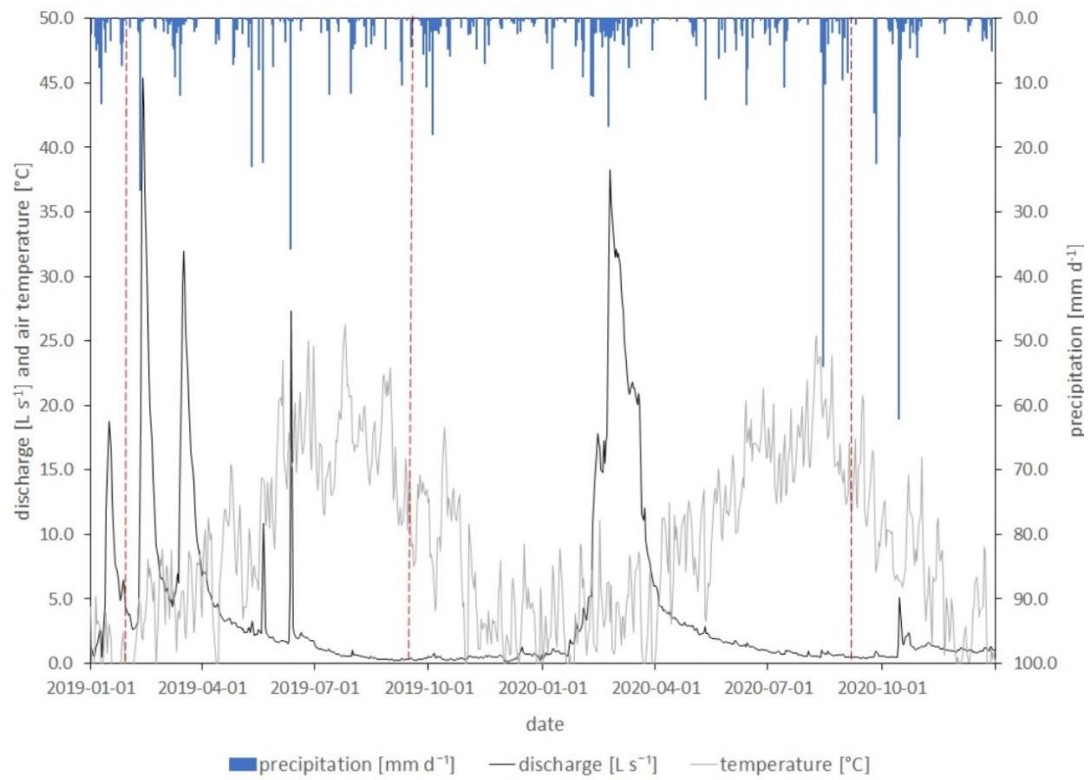

**Figure 2:** Daily precipitation (mm d⁻¹), daily average discharge (L s⁻¹), daily average air temperature (°C) for the Schäfertal
stream during the measurement campaign period. Red dashed lines mark campaign dates.

### 2.2.1 Water inflows to the stream

### 2.2.1.1 Water balance of stream sections measured by tracer dilution tests

Consecutive salt tracer dilution tests were applied to quantify gross gains, gross losses and net change in water flow following
Payn et al. (2009). The tracer tests were performed in January 2019, September 2019 and September 2020 at six locations
defining five stream sections with lengths between 130 and 250 m. The total length of the stream sections studied was 955 m.
The sodium chloride tracer was prepared in the lab and diluted in 7 l of stream water prior to injection. In the two 2019
campaigns, 500 g of tracer was used for each injection while in the 2020 summer campaign, 100 g of tracer was used. The
tracer was injected 10 m upstream of each observation point, giving a mixing length of 20 to 30 times the stream's width.
Tracer breakthrough was measured at 5-second intervals using individual in situ conductance loggers (Eijkelkamp CTD-
Divers) fixed in a central position in the stream. The tracer was injected working consecutively upstream from the observation
point with a time interval of 30 min to ensure that the breakthrough curves of consecutive injections did not overlap. The
measured time series for specific conductance were converted to sodium chloride concentrations using a linear regression (with

the intercept fixed at zero) for each individual data logger based on four data points with known sodium chloride concentrations (measured in the laboratory). The background specific conductance was subtracted from the time series prior to conversion. For each tracer breakthrough concentration [mg L$^{-1}$], the time series was summed up to the mass flux [mg L$^{-1}$ s$^{-1}$]. A known injected tracer mass [mg], makes it possible to derive discharge [L s$^{-1}$]. Following Payn et al. (2009), we quantified breakthrough of the injection at each observation point in relation to the breakthrough of the injection at the observation point immediately upstream, assuming that the net change in discharge is the sum of gross gains and gross losses along the stream. The net change for a given section is calculated as the difference between upstream and downstream discharge measurements. Gross loss for each section is derived at the downstream observation points from mass recovery analysis of the upstream injected tracer and upstream discharge. Gross gain for each section is derived from the difference between net change and gross loss. Note that, in the 2019 summer campaign, it was not possible to measure upstream injection breakthrough for the uppermost two sections because flow velocity was too low for breakthrough to be measured in the allotted time. In summer 2020, this was the case for the uppermost section only. Here, only net changes could be quantified.

### 2.2.1.2 Groundwater discharge investigated by radon ($^{222}$Rn) measurements

The natural radon ($^{222}$Rn) activity in the stream water was used in addition to the salt tracer investigations to provide insight into both the spatial distribution and the quantity of groundwater discharge into the Schäfertal stream along the stream section. Radon is an excellent tracer for investigating groundwater-surface water interactions (Adyasari et al. 2023). Longitudinal stream radon measurements allow (i) localization of groundwater discharge zones and (ii) calculation of radon mass balances within defined sections of the stream and subsequently groundwater discharge into the stream. A crucial parameter for this method is the rate of radon degassing from the stream to the atmosphere, which is primarily dependent on the stream turbulence, i.e., on stream geometry, streambed roughness and stream flow velocity (Genereux and Hemond 1992, Raymond et al. 2012). A number of experimental and empirical methods are available to estimate radon degassing from a stream. A detailed discussion of the approach applied specifically to the Schäfertal stream is provided in Schubert et al. (2020). Furthermore, Raymond et al. (2012) have published a comprehensive review of scaling gas transfer velocities in streams and small rivers.

Radon mapping along the Schäfertal stream was carried out during the low flow measurement campaigns in January 2019 and September 2020. During each campaign, stream-water samples were taken from six locations distributed (roughly) equidistantly along the stream, thus subdividing the study reach into five sub-sections (*cf.* Figure 6). The radon sampling points were identical to the salt tracer measurement points. To determine the lumped radon groundwater and tile drainage endmember (hereinafter referred to as "lumped radon groundwater endmember") water samples were taken from three groundwater wells located adjacent to the stream and from two subsurface tile drains that discharge into the stream within the section located furthest upstream. Radon measurements were carried out on-site immediately after sampling using a mobile radon-in-air monitor (RAD7) as described by Schubert et al. (2006). Groundwater discharge localization and quantification were performed using a mass-balance approach in the implicit finite element model FINIFLUX2.0, which is described in detail in Frei and

Gilfedder (2015). FINIFLUX numerically solves the mass balance equation for stream $^{222}$Rn at the reach scale by using a Petrov-Galerkin Finite Element scheme fitting modelled radon to the measurement radon results. The model input parameters include (i) the length of the investigated sub-section, (ii) its mean width and mean depth, (iii) the discharge of the stream, as well as the $^{222}$Rn activities of (iv) the stream water specific for the sub-section and of (v) the overall lumped radon groundwater endmember. Hyporheic exchange is also allowed for by the model (based on the physical characteristics of the hyporheic zone and used as an optimisation parameter). The model accounts for radon losses such as (vi) degassing using a water-air exchange coefficient (k, specific for each stream sub-section) and (vii) the first order decay constant for $^{222}$Rn.

### 2.2.2    Stream water, groundwater and stream sediments chemistry

Longitudinal water-quality measurements (n=11) were carried out during the three campaigns at intervals of approximately 100 m, starting at the catchment outlet gauging station (in the "Messgarten") using a YSI 610 multiparameter probe ($O_2$, pH, electric conductivity) and a TRIOS ProPS-UV sensor with an optical path length of 2 mm ($NO_3^-$). Probes were calibrated on the day of measurement. Further details on accuracy and precision detection limits are given in Rode et al. (2016). Additional grab samples were taken for the measurement of SRP and total dissolved phosphorus (TDP) using standardised methods. The pore size of filters were 0.22 µm. TP and SRP were measured using the ammonium molybdate spectrometric method (DIN EN ISO 6878, 2004). The detection limit was 0.005 mg P $L^{-1}$. We operationally define the difference between TDP and SRP as dissolved organic phosphorus (DOP), although this fraction may contain some inorganic phosphorus species.

To elucidate SRP concentrations in potential source zones and possible redox-mediated mobilization, SRP and dissolved Fe from groundwater wells were measured during the September 2020 campaign. Dissolved Fe concentrations were measured by ICP-OES (Perkin Elmer 7300 DV). The detection limit was 0.01 mg $L^{-1}$. Additionally, SRP and dissolved Fe were measured in gaining groundwater, streambed pore water and the stream water. Groundwater samples were taken from six wells near the stream using a peristaltic pump. Streambed pore water samples were taken at two points, one located 400 m from the head of the study reach ('upstream station'), and the other located 900 m from the head ('downstream station' or 'outlet'). Gaining groundwater was sampled at the upstream station. Additionally, samples for $^{14}$C-DOC were taken from stream water and from stream-sediment leachate from both the upstream station and outlet as well as from gaining groundwater at the upstream station.

Pore water and gaining groundwater were sampled using PTFE piezometers with a diameter of 10 mm and a screen length of 80 mm. The piezometers were placed in the streambed sediments with the help of a solid metal rod, either at a depth of 7 cm below the streambed surface (pore water), or at the bottom of the sediment at a depth of 15 cm (gaining groundwater). Samples were drawn through a PTFE tube with a syringe and filtered with 0.45 µm cellulose acetate filters on site before being transported in gas-tight flasks without headspace and cooled in the laboratory.

Streambed sediment samples were taken at both stations using a shovel from depths of 5 cm. In the laboratory, 25 g of sediment was slurried with 150 mL of deionized water and incubated for 24 h in an overhead shaker at 20°C in the dark. The pH ranged between 5.0 and 6.1 at the end of incubation. After centrifugation (5250$g$, 15 min) the supernatant was filtered (Whatman

GF/F, pre-combusted for 4 hours at 500°C) and the DOC in the water samples was processed for radiocarbon analysis as described previously (Tittel et al., 2013). Radiocarbon quantities were analysed by accelerator mass spectrometry (AMS) at the Poznan Radiocarbon Laboratory (Poland). The results refer to the oxalic acid II standard and were corrected for fractionation (Stuiver et al., 1977).

## 3 Results

### 3.1 Observed discharge and stream SRP concentrations

During the January 2019 campaign, the stream discharge at the outlet was 5.35 L s$^{-1}$. This campaign was carried out in the recession phase of a discharge event (Fig. 2) with no visible active surface flow to the stream. By contrast, the summer campaigns were carried out under strong drought conditions and the stream discharge measured at the outlet was only 0.26 L s$^{-1}$ (Sep. 2019) and 0.51 L s$^{-1}$ (Sep. 2020) (see Figure 2). During the two summer campaigns, SRP concentration at the outlet (0.068 mg L$^{-1}$ in Sep. 2019 and 0.041 mg L$^{-1}$ in Sep. 2020) was higher by a factor of 4 to 8 than in the winter campaign (0.009 mg P L$^{-1}$ in Jan 2019). SRP concentrations displayed very constant longitudinal behaviour with only a very slight increase upstream to downstream from 0.008 mg P L$^{-1}$ to 0.009 in Jan. 2019 (Figure 3). We monitored more pronounced increases along the stream in Sep. 2019 from 0.024 mg P L$^{-1}$ to 0.068 mg P L$^{-1}$ and in Sep. 2020 from 0.022 mg P L$^{-1}$ to 0.040 mg P L$^{-1}$. Total dissolved P (TDP) was 0.040 mg L$^{-1}$ higher than SRP during both 2019 campaigns but only 0.001 mg L$^{-1}$ higher in Sep. 2020. The strongest increase in SRP concentrations was found at a distance of approximately 200 m from the head of the study reach during summer campaigns, whereas the highest concentrations in Jan. 2019 were observed at a location approximately 400 m from the head of the reach (Figure 3). Nitrate concentrations tended to increase from upstream to downstream during both summer campaigns up to 3.81 mg N L$^{-1}$ (Sep. 2019; mean 3.59 mg N L$^{-1}$) and 4.48 mg N L$^{-1}$ (Sep.2020; mean 4.50 mg N L$^{-1}$). A much higher mean $NO_3^-$ concentration of 12.7 mg N L$^{-1}$ was recorded during Jan. 2019.

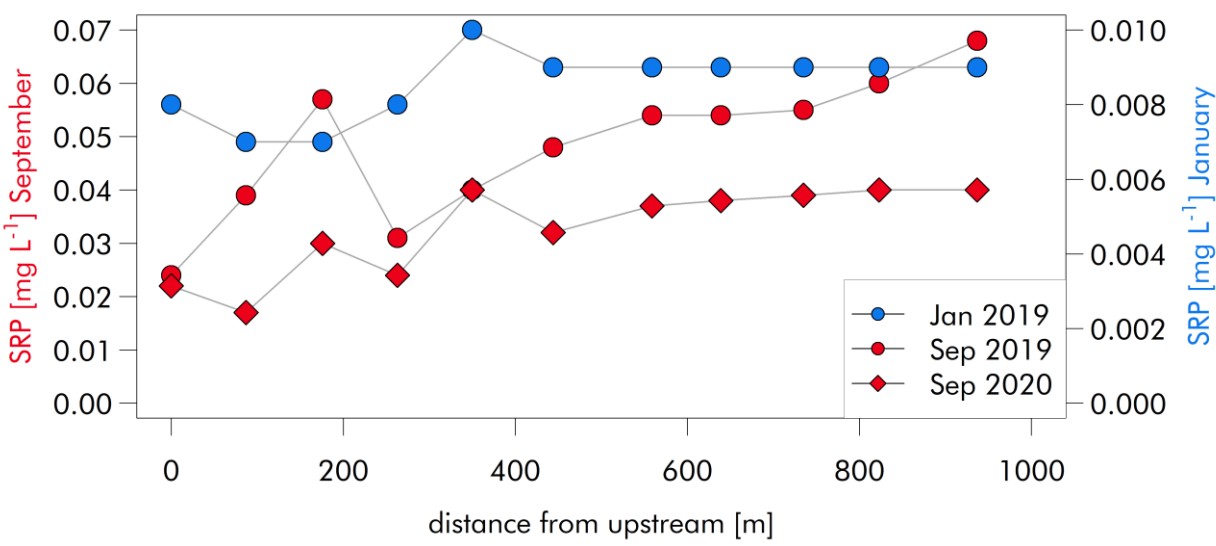

**Figure 3:** Longitudinal SRP concentration profiles for the three measurement campaigns in January 2019, September 2019 and September 2020.

### 3.2 Observed longitudinal water and SRP fluxes

The salt tracer dilution tests produced a distinct, non-uniform distribution of increases in discharge. In the Jan. 2019 campaign, the two uppermost stream sections (0 to 460 m) cumulatively gained 44% of the discharge observed at the catchment outlet gauging station. There was no further increase in discharge in the three downstream sections, while the lowest stream section appeared to have significant losses. In the two September campaigns, a high proportion of the increases in discharge occurred in the second section alone (210 m to 460 m from upstream). This section could account for 52% (2020) to 89% (2019) of the discharge observed at the outlet of the catchment.

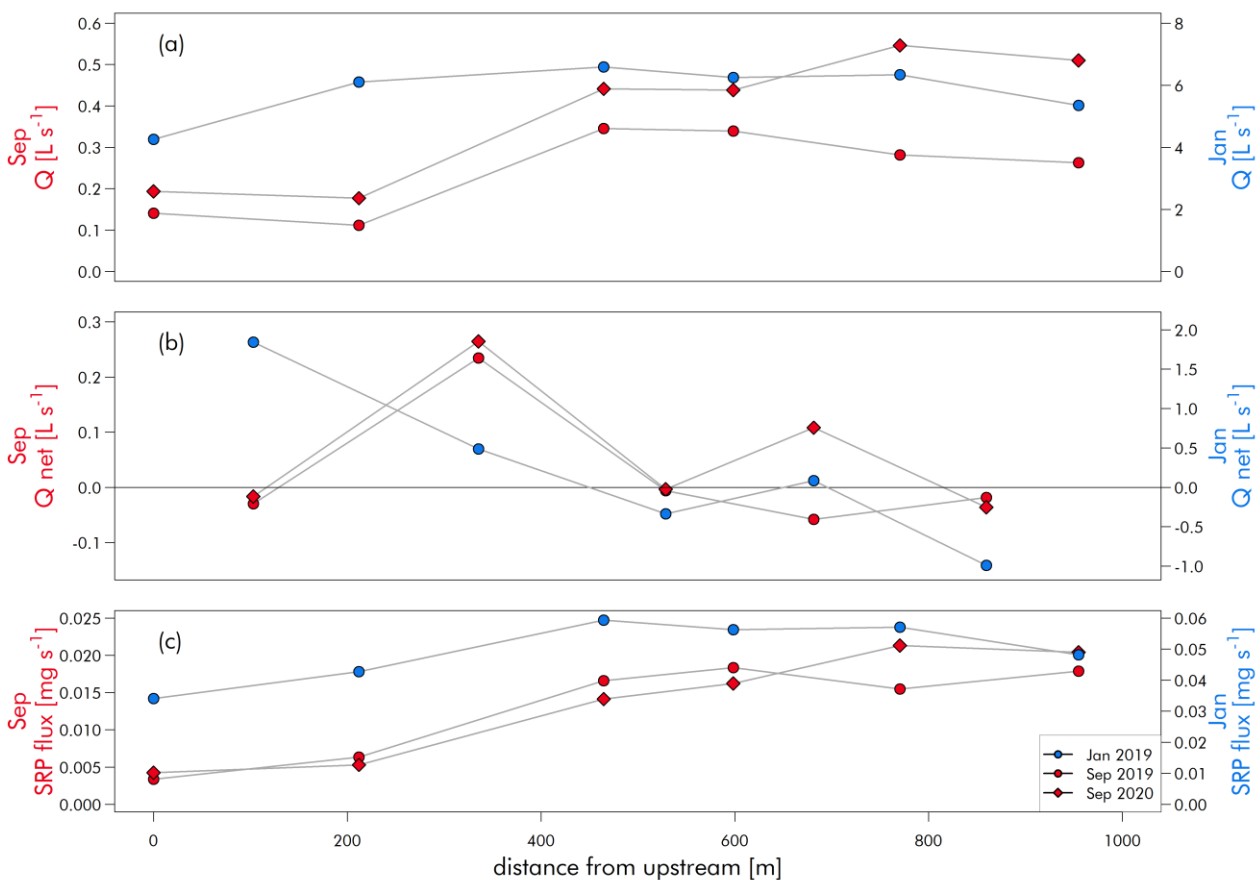

**Figure 4:** Discharge, net discharge gains and losses, and SRP flux along the stream during the three sampling campaigns.

The SRP flux at the catchment outlet in the winter campaign (0.048 mg s⁻¹) was double the flux observed during the two September campaigns (0.018 and 0.028 mg s⁻¹, respectively). The spatial pattern of discharge along the stream largely translated to the pattern observed for SRP flux. In January 2019, the two upstream sections gained 52% of the SRP flux observed at the outlet, while the downstream section lost a small amount of water and SRP. In the September campaigns, just one stream section gained 43% (2020) and 57% (2019) of the SRP flux observed at the outlet. In 2020, we observed a further increase in the lowest three sections of 30%, which had not been observed in 2019.

### 3.3 Groundwater discharge investigated by ²²²Rn measurements

The two radon sampling campaigns ("winter": Jan. 2019 and "summer": Sep. 2020) resulted in almost identical radon concentration distribution patterns along the study reach, with high radon concentrations in the two upstream sub-sections and concentrations declining exponentially at approximately similar rates in the three downstream sub-sections (Figure 5). The lumped radon groundwater endmember (as determined from samples taken from three groundwater wells and two tile drains)

averaged $23.2 \pm 1.14$ kBq m$^{-3}$. In the summer campaign, the very low water level at the sampling point furthest upstream made it necessary to dig a small depression in the streambed sediment for stream-water sampling. This disturbance to the streambed is highly likely to have resulted in a minor but locally significant preferential groundwater discharge pathway, leading to a radon concentration in this particular sample that can be assumed to be closely representative of pure groundwater. Thus, the

value detected here was not considered representative of stream water at this location (illustrated as dashed line in Figure 5). Although the radon distribution patterns of the winter and summer plots are roughly comparable, the concentrations observed during the winter campaign were significantly higher than in summer. This general difference is thought to result from the differences in hydrological gradients. The summer campaign was conducted under strong drought conditions, implying low groundwater levels and low groundwater discharge to the stream. By contrast, the winter campaign was characterized by high

groundwater levels generating normal baseflow (incl. interflow from shallower stores and/or drains, i.e., from the soil matrix) into the stream.

Groundwater discharge rates were calculated based on the lumped radon groundwater endmember and the stream radon activities displayed in Figure 5 using the FINIFLUX model. The results for the five stream sub-sections consistently showed that the majority of groundwater discharge occurred in the second upstream section. The exponential decrease in radon

concentration in the following three downstream sections suggests that degassing is the dominant radon sink with little or no contribution from radon sources such as groundwater discharge or hyporheic exchange.

The section-specific quantification of groundwater discharge was more difficult than their localization, due to considerable uncertainties in parameterizing radon degassing (i.e., in defining a degassing coefficient k [m d$^{-1}$]) for the FINIFLUX model in such a small stream. The uncertainties of the model results are mainly correlated to the uncertainty of the applied degassing

coefficient (for a detailed discussion see Schubert et al. 2020). The k values that resulted from our FINIFLUX model (for details see Frei and Gilfedder, 2015) for the summer and winter campaign were 4.2 m d$^{-1}$ and 2.3 m d$^{-1}$, respectively. In order to, as a first step, roughly assess the range of reasonable k values, we calculated k for the stream using three equations introduced by Raymond et al. (2012) (their Eqs. 1, 2 and 7 there). The resulting k values ranged from 1.1 m d$^{-1}$ to 3.0 m d$^{-1}$ (their Eq. 2 and Eq. 7 respectively) for the summer campaign and from 19 m d$^{-1}$ to 29 m d$^{-1}$ for the winter campaign (their Eq.

1 and Eq. 7, respectively). This shows that the equations of Raymond et al. (2012) give in our case reasonable results only for the summer campaign. Using the above estimates from Raymond et al. (2012) for the winter campaign with FINIFLUX resulted in an excess of groundwater compared to the actual water balance. One possible reason for the observed discrepancies in winter may be that the empirical equations introduced by Raymond et al. (2012) are generally based on data collected in rivers much larger than the small headwaters we examined in this study.

For estimating how sensitive our modelled groundwater discharge rates are to the degassing coefficient that was finally considered most reasonable for our FINIFLUX model setup, we calculated a range of uncertainties for each campaign. For this purpose, we increased/decreased the degassing coefficient applied in FINIFLUX by $\pm 25$ % and ran several individual model calculations.

The resulting cumulative groundwater discharge rates for the winter campaign was 0.75 L s$^{-1}$ ranging from 0.42 to 1.13 L s$^{-1}$

(mean 0.75 L s$^{-1}$) with the higher discharge rates associated with + 25 % values for the degassing coefficient. Hence, a ±25 % uncertainty in the applied degassing coefficient corresponds to a relative uncertainty in the modelled groundwater discharge ranging between ca. -44 and +51 %. Assuming 0.75 L s$^{-1}$ as cumulative groundwater discharge rate within the study reach for the winter campaign suggests that about 63 % of all water entering the stream within the reach (water balance 1.2 L s$^{-1}$) was groundwater.

For the summer campaign the groundwater discharge was modelled to be 0.5 L s$^{-1}$. The ±25 % variation resulted in groundwater discharge estimates ranging from 0.20 to 0.47 L s$^{-1}$. Note that the +25 % run resulted in a discharge value (0.47 L s$^{-1}$) that is slightly lower than the result of the ±0 % run (0.5 L s$^{-1}$). The reason for this is that the model fit was slightly better for the down-stream reaches in the +25 % scenario. Hence, for the summer campaign the ±25 % variance corresponds to a relative uncertainty in the modelled groundwater discharge ranging between -60 % and -6 %. However, the modelled groundwater

discharge of 0.5 L s$^{-1}$ is physically impossible as the water balance of the stream section was lower (0.31 L s$^{-1}$). The difficulty arises from low stream discharge during the summer campaign (0.51 L s$^{-1}$), which lead to a very shallow water level (only centimetres) and a very low flow velocity of the stream water (around 0.07 m s$^{-1}$). This resulted in intense radon degassing from the stream along its flow path as well as a rather high uncertainty when quantifying the degassing constant. Even though the FINIFLUX model can generally allow for such high radon loss by degassing, it reaches its limits for the modelling of the

groundwater discharge quantities during the summer campaign. A physically possible cumulative groundwater discharge rate calculated based on the radon data for the summer campaign results from using the -25 % value for k for calculating the degassing (i.e., 0.2 L s$^{-1}$). The most plausible assumption for the summer campaign based on the radon data is a cumulative groundwater discharge rate between 0.2 and 0.3 L s$^{-1}$, suggesting that nearly 100 % of the water gained by the stream during the summer campaign along the study reach was derived from groundwater.


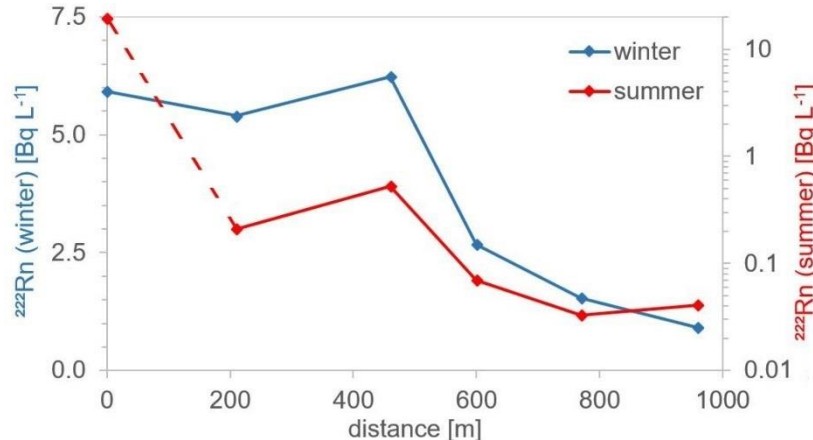

**Figure 5:** [222]Radon concentration patterns along the stream sections during winter (Jan 2019) and summer (Sept. 2020) campaigns; the dashed line indicates that the most upstream "summer" sample cannot be considered representative of stream water (see text above)

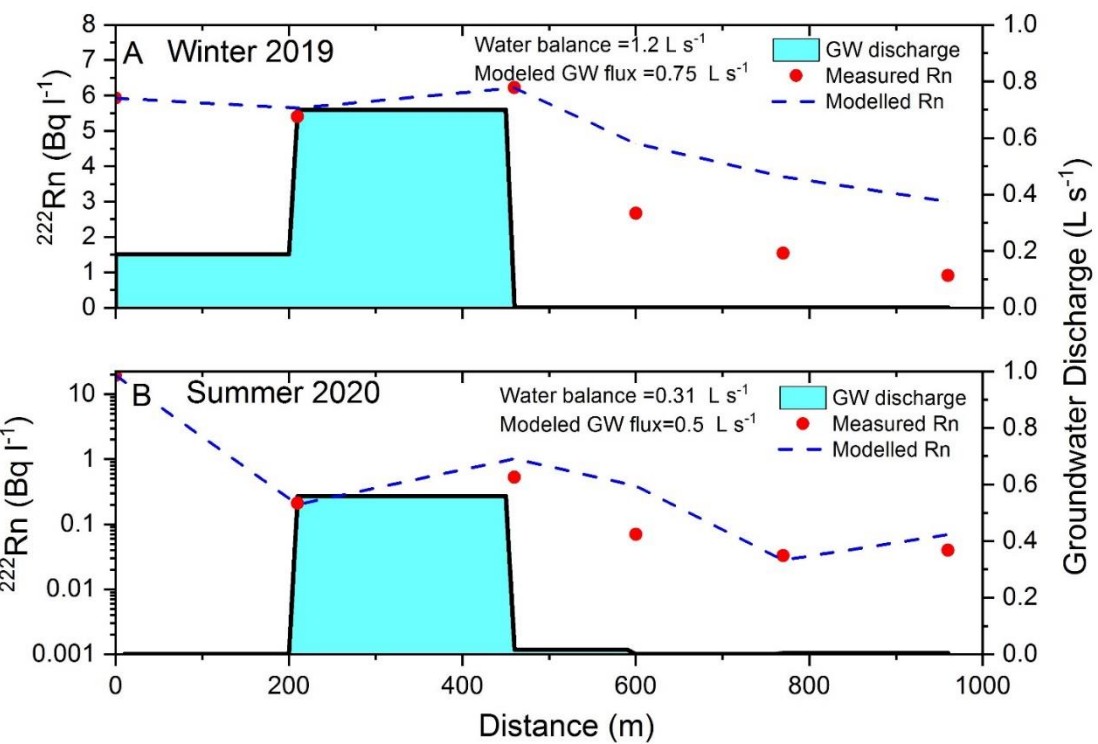


**Figure 6:** [222]Radon concentrations in the stream water (measured and modelled) and groundwater discharge levels during A the winter (Jan. 2019) campaign and B the summer (Sep. 2020) campaign; groundwater discharge during the winter campaign includes drain water discharge; note that the most upstream "summer" sample was not considered representative of
stream water (see Figure 5)

### 3.4 Assessing potential sources of SRP in summer baseflow (Sep. 2020)

Under summer low-flow conditions in Sep. 2020, we conducted a systematic survey of potential sources for the SRP entering the stream: streambed sediments and sediment pore water were sampled at the upper observation point of the section with the
highest gains (400 m from the top of the study reach) and at the downstream discharge gauging station located at the outlet. Further samples of groundwater were taken from the surrounding wells and directly from the deeper part of the streambed at the upstream station. SRP, Fe, DOC and $NH_4^+$ concentrations, ratio of SRP/DOP, where we define dissolved organic phosphorus (DOP) as TDP minus SRP, and radiocarbon age were used to compare the characteristics of these potential source waters with those of the stream water.

Electric conductivity in stream water differed only slightly between the stations (upstream 0.24 mS cm$^{-1}$, downstream 0.25 mS cm$^{-1}$) and pH was 7.4 at both stations. Stream SRP concentration ranged between 0.029 and 0.041 mg L$^{-1}$. SRP contributed 95 % of dissolved (inorganic and organic) P (Tab. 1). The fraction of DO-P was therefore insignificant. In the samples from the groundwater wells and gaining groundwater, SRP concentrations were in the same order of magnitude (0.012 – 0.068 mg L$^{-1}$) and SRP also constituted the dominant fraction of TDP (92 ± 15 %, mean ± SD). Low Fe and NH$_4^+$ concentrations in the

groundwater indicate oxidizing groundwater conditions (see Table 1). Sediment porewater concentrations differed substantially between stations. At the outlet, an elevated SRP porewater concentration exceeding 0.600 mg L$^{-1}$ was recorded, whereas other P fractions were insignificant. At the upstream station, low sediment porewater levels of SRP (0.022 mg L$^{-1}$) similar to those in the stream were found, but the dissolved P concentration (0.312 mg L$^{-1}$) was more than tenfold higher. The SRP/DOP ratio in the sediment was 0.078 and in stream water ~0.935. It is unlikely that turnover in the stream with such a

short travel time reverses the ratio of sediment-borne P. It could therefore be concluded that the sediment in the upper part of the stream was unlikely to be a significant source of stream P.

At the outlet concentrations of Fe, DOC, dissolved P and NH$_4^+$ were higher in the sediment pore water than those in the stream (Tab. 1), indicating iron–reducing conditions and anaerobic decomposition of organic matter. At both stations, the accumulation of Fe, dissolved P and NH$_4^+$ in the sediment was consistent with only diffusive fluxes and a quantitatively

insignificant source of porewater solutes to the stream.

**Table 1:** Concentration of SRP, TDP, Fe, DOC, NO$_3^-$, NH$_4^+$ and $\Delta^{14}$C-DOC at upstream and downstream stations (outlet) in different stream compartments and in groundwater wells, sampling campaign Sep. 2020

| Compartment | Station | SRP [mg L$^{-1}$] | TDP [mg L$^{-1}$] | Fe [mg L$^{-1}$] | DOC [mg L$^{-1}$] | NO$_3^-$ [mgN L$^{-1}$] | NH$_4^+$ [mg N L$^{-1}$] | $\Delta^{14}$C [‰] |
|---|---|---|---|---|---|---|---|---|
| Stream | upper | 0.029 | 0.031 | 0.159 | 2.05 | 2.32 | 0.06 | 127±4 |
| | outlet | 0.041 | 0.040 | 0.027 | 2.48 | 3.58 | 0.05 | -178±3 |
| Sediment pore water | upper | 0.022 | 0.312 | 2.880 | 5.46 | 2.11 | 0.13 | - |
| | outlet | 0.654 | 0.630 | 26.40 | 9.12 | 0.75 | 3.55 | - |
| Streambed sediment leachate | upper | - | - | - | - | - | - | 72±4 |
| | outlet | - | - | - | - | - | - | 6±5 |
| Gaining groundwater | upper | 0.040 | 0.039 | 0.805 | 1.33 | 0.68 | 0.04 | -246±3 |
| Groundwater wells, mean values | n=6 | 0.038 | 0.043 | 0.611 | 1.20 | 9.27 | 0.119 | - |

Stream DOC radiocarbon ratios differed dramatically between the two sampled stations. At the upper station, the DOC was enriched in radiocarbon ($\Delta^{14}$C 127 ‰), meaning that the organic carbon was young, containing post-bomb carbon that had been fixed photosynthetically after 1950. At the outlet, by contrast, stream water DOC exhibited a negative $\Delta^{14}$C of -178 ‰, corresponding to a conventional radiocarbon age (CRA) of ~1500 years B.P. This signature was much closer to the upwelling

gaining groundwater DOC (-246 ‰, CRA 2200 years B.P.) than to the DOC signature for the upper station. However, in

experiments performed on incubated streambed sediment samples from both stations, young DOC was seen to be released ($\Delta^{14}$C 6 to 72 ‰).

## 4 Discussion

### 4.1 Main pathways of SRP transfer into streams

In all three sampling campaigns results of the longitudinal SRP concentration and flux analysis indicate a distinct zone where most of SRP enters the stream. The radon analysis showed that a large part of the water entering the stream in the winter campaign and all of the gained water in the summer campaign 2019 can be explained by inflowing groundwater or tile drain water. Combining the $^{222}$Rn-based groundwater (incl. tile drain water) inflow and it's uncertainties (chapter 3.3) with the SRP concentration observed in groundwater wells (Table 1) yields a SRP flux of 0.008 - 0.064 mg s$^{-1}$ (mean 0.029 mg s$^{-1}$) for winter

2019 and 0.004 - 0.064 mg s$^{-1}$ (mean 0.012 mg s$^{-1}$) for summer 2019. The observed SRP fluxes at the catchment outlet (0.048 mg s$^{-1}$ in winter 2019 and 0.018 mg s$^{-1}$ in summer 2019) are within the range of the estimated incoming SRP flux. This also suggests that SRP fluxes were not significantly buffered by in stream and hyporheic processes that may play a role under summer low flow conditions. This is consistent with the results of Bernot et al. (2008), who observed very low SRP uptake in small agricultural streams with comparably low SRP concentrations. We therefore argue that the groundwater fluxes inferred

from the $^{222}$Rn data and SRP groundwater concentration can explain the SRP flux observed at the catchment outlet. This means that groundwater inflow could be a dominant pathway for SRP enter to the stream.

Our detailed hydrochemical and radiocarbon analysis in summer 2020 reveals that the SRP and TDP concentrations and DOC radiocarbon age of the stream water at the outlet compare most closely to the groundwater entering the stream in the upper

stream section. Sediment pore water quality differed from that of the gaining groundwater and stream water, with much higher TDP, $NH_4^+$ and DOC concentrations. Thus, gaining groundwater did not interact significantly with the streambed pore water and was probably transported by preferential flow paths, such as the tile drains. It is known that a fine-grained streambed with low hydraulic conductivity favours preferential flow paths so that largest part of incoming groundwater flux is channelled to a few distinct locations (Schmidt et al. 2006). Sediment pore water, in particular that sampled at the outlet, exhibited high

concentrations of TDP, Fe and $NH_4^+$ but low concentrations of $NO_3^-$ along with young DOC radiocarbon ages, suggesting that reductive conditions predominate for mobilization of Fe and TDP (Smolders et al. 2017). This indicates that sediment was a source of P to the stream, but that transport was quantitatively limited and likely of diffusive and not advective nature only. However, there was no further interaction between the stream water and the sediment pore water at downstream points along the stream length, instead, the stream water retained the signature introduced in the upstream section through the gaining

groundwater. We should note that the stream water sampled at the uppermost station showed no signature from the gaining groundwater.

The young radiocarbon age of the upstream DOC suggests its source lies in shallow organic rich sediments, such as riparian wetland soils. In contrast, the presence of aged DOC in groundwater, here with a radiocarbon age of more than 2000 years BP, is typical of low flow periods (Schiff et al. 1997, Tittel et al. 2022). The groundwater is unlikely to be that old; the DOC

may have been mobilized from the aquifer matrix recently by hydrolysis of historic organic carbon in the soil. The limited influence of shallow porewater in the upstream, strongly gaining section is largly due to the different P species in porewater compared to that observed in the stream water downstream. The limited influence of the streambed SRP sources is supported by i) distinct water and SRP flux gaining in the upstream section but not in the downstream section, ii) hydrochemical similarity of stream water and gaining groundwater including tile drains, and iii) hydrochemical dissimilarity of sediment pore water and

the stream water.

A similar argumentation holds true for potential lateral inputs from shallow anoxic riparian wetlands as suggested by Dupas et al. (2017a). Given the lack of evidence of significant water and SRP fluxes apart from the upstream section and the hydrochemical similarity of stream water and gaining groundwater we argue that water from anoxic riparian wetlands were not a significant source for SRP to the Schäfertal stream during the low flow sampling campaigns.


## 4.2 Explaining seasonal variability and pathways of SRP by integrating current observations with measurements from previous studies

The long history of studies on water quality in the Schäfertal catchment (see Ollesch 2008) allows us to integrate the results of the three sampling campaigns with the wider context of variability in seasonal and discharge dependent SRP concentrations.

Additionally, earlier work allows a comparison with tile drainage water SRP concentrations, which may play a role under high flow conditions.

Stream water quality observations from previous assessments cover the years 1999 to 2010. We should note that within this time series there is a small, but significant, increasing trend in SRP concentrations in stream water (Mann-Kendall test, average increase 0.65 µg L$^{-1}$ a$^{-1}$). Previous SRP concentrations are comparable to those presented here (Figure 7). Mean groundwater

SRP concentrations in previous assessments of 0.072 mg P L$^{-1}$ (Figure 7) were higher than those sampled in September 2020. The number of wells sampled in past studies was higher than the six wells located close to the stream that were the focus of the September campaign. When the average SRP concentration (34.1 µg P L$^{-1}$) were examined for these six wells only, there was a good agreement between historic values and those obtained in the present study. Water samples from tile drains in 1999 and 2004 showed somewhat low SRP concentrations (0.011 mg P L$^{-1}$) with a low temporal variability (Std. dev. of 0.006 mg

P L$^{-1}$) (Figure 7), although most of the drain samples (73%) were taken under colder high-flow conditions between January and May.

Stream SRP concentrations displayed clear seasonality, with the highest concentrations occurring under summer low flow conditions. The resultant concentration-discharge relationship (C = a Q$^b$) yields an exponent b of -0.24 (SE 0.045) and thus indicates a mild dilution pattern (Figure 7). The three sampling campaigns in this study capture the typical range of discharge

conditions and associated SRP concentrations very well (Figure 7). The observed mild dilution pattern is in good agreement with an assessment of concentration dynamics of dissolved $PO_4^{3-}$ in over 700 catchments across Germany: for example Ebeling et al. (2021) state a mean exponent b of -0.22 and a dominance of dilution patterns in German catchments.

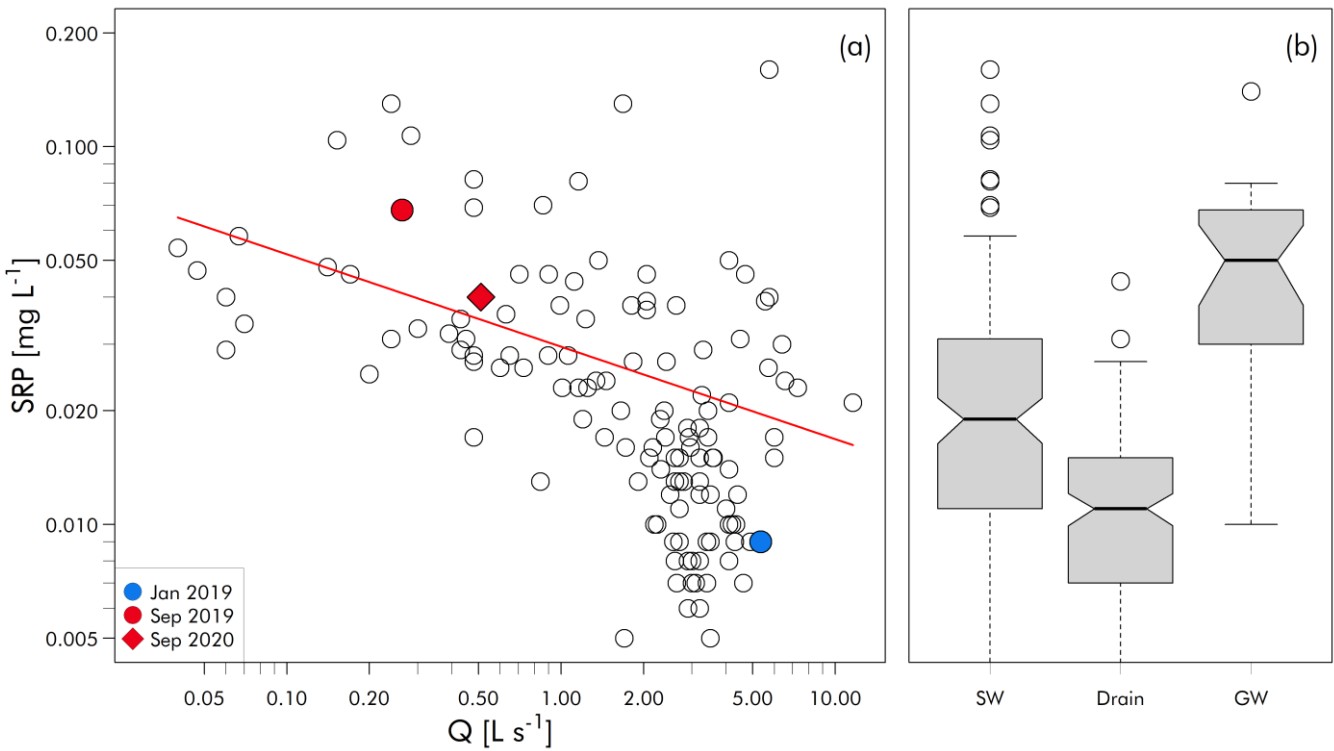


**Figure 7:** Long-term distribution of concentration discharge relationships for stream SRP at the catchment outlet (left), Box-Whisker plots of concentrations in stream water (SW, n=146), tile drains (Drain, n=138), and groundwater (GW, n=25) (right). Data are from Kistner (2007) (Drain), Dupas et al. (2017b) (SW), and UFZ TERENO monitoring, unpublished (GW) (see Supplement).


The SRP source partitioning provides strong evidence that groundwater inflow into the stream is the major contributor of SRP in the Schäfertal catchment throughout the year. The groundwater discharge dominates stream concentrations under summer low-flow conditions, with stream and groundwater concentrations observed in the same range. Under higher flow conditions other, younger, water components dilute the broadly constant groundwater signature. While there is evidence for reductive

mobilization of SRP within the streambed-sediment pore water, this source does not contribute significantly to exported SRP. The seasonality of SRP concentration is thus not predominantly driven by in-stream biogeochemical processes and therefore is caused by the seasonally-variable proportion of groundwater to total stream discharge. Therefore they are highest when groundwater dominates stream discharge under summer low-flow conditions. This dominance of groundwater SRP in stream

water SRP during low-flow periods was also recorded by Jarvie et al. (2008) in some rural catchments in the UK. Findings from Holman et al. 2008 in the UK also suggest that groundwater can be an important contributor to surface water P especially when it is dominating discharge. It also matches the dilution patterns of dissolved P observed in German catchments (Ebeling et al. 2021). Although groundwater SRP concentrations are moderate in our study, they are well above the critical surface-water threshold for eutrophication of 0.02 to 0.03 mg P L$^{-1}$ (Corell, 1998, King et al. 2014). These findings apply only to typical hard-rock mountain ranges, and groundwater concentrations may differ in other geological settings. Geogenic SRP concentrations can reach even higher levels where organic-matter content is higher or redox processes are more prevalent in soils and riparian wetlands, as is typically the case for peatlands. The high mobilization of P under reducing conditions may increase its bioavailability; however, it may also increase its loss from soils, particularly in the toe-slope profile (Shaheen et al. 2021).

Furthermore, we show that P release from stream sediments was not a major source of stream water P during summer low-flow condition. This finding is consistent with the results of the review by Simpson et al. (2021) who found that on average there was a negative net phosphate exchange potential, meaning that sediments predominantly have a potential to remove P from the water column. In general, stream sediments often have potential to exchange P with the water column and can buffer (retain or release) SRP (Houser 2003, Whiters and Jarvie, 2008, Weiglhofer et al. 2017, Simpson et al. 2021). This phosphate exchange potential can vary depending on seasonally fluctuating environmental factors (e.g., temperature, light, stream discharge, redox and sediment inputs) (Simpson et al. (2021). Recent findings suggest that stream sediments can act as a source when P loading is elevated, the SRP/Fe ratio is high and DO concentrations are low (van Deal et al 2021a). Although we observed anoxic conditions in stream sediments, the rates of diffusive transport are too low to release any significant quantity of P into the stream water. P releases from stream sediments tends to be more significant in slow-flowing lowland streams with considerable legacy P from point source inputs (van Dael et al. 2020). Such experimental evidence of P release from stream sediments is still rare, but new modelling approaches may help to assess potential P losses from stream sediments to the water column (van Dael et al. 2021b).

### 4.3 Potential sources of SRP in groundwater

Groundwater in the studied catchment exhibit elevated concentration of SRP (see chapter 3.4 and 4.2). Potential sources of this SRP in groundwater include agricultural land use and the geologic parent substrate. The mean SRP concentrations found in the groundwater (0.05 mg P L$^{-1}$, Figure 7) were slightly above mean background concentrations of groundwater from Paleozoic greywacke and Devonian shale of around 0.025 mg P L$^{-1}$ but still within the observed range of 0.01 to 0.10 P L$^{-1}$ (Wriedt et al. 2019). Our findings suggest that SRP concentration in the saturated zone is controlled by oxidizing conditions in the upper groundwater, and that SRP losses through seepage are largely buffered. This is in line with the results of Wriedt et al. (2019) who could not find general differences in groundwater SRP concentrations between land use types (arable land, grassland and forest) for these geological units. The study site is characterized by loamy soils which in general display low susceptibility to the preferential flows that can be critical for subsurface SRP losses (Stamm 1998, Simard et al. 2000). In a

typical agricultural soil of the study catchment, TP concentrations were elevated in the $A_p$ horizon (778 mg kg$^{-1}$) but showed a dramatic drop to a mean value of 193 mg kg$^{-1}$ in the B and C horizon below the plough pan in 40 cm depth, indicating a high

P sorption capacity of the soil. The loamy soils in the study catchment show also a distinct reduction in $C_{org}$ concentration with increasing soil depth. This suggests that P is unlikely to be leached from the $A_p$ horizon to a greater extent into deeper layers of the studied soil (Kistner 2007). This is consistent with Leinweber et al. (1999) who found that fine-textured soils have a much lower potential for P leaching than sandy or organic soils, unless there are high P accumulations in the soil (Reid et al. 2012). In addition, the study catchment showed low levels of organic fertilizer (manure) application which, under certain

conditions, can increase SRP losses to streams (McDowell et al. 2005, King et al. 2014). Although our analyses show that soil leaching of SRP in the study catchment is possibly low, we cannot completely rule out SRP leaching from agricultural soils to groundwater due to slightly higher mean given SRP concentrations in groundwater compared to the geological background values. Currently enriched P in agricultural soils (Schachtschabel et al. 1992, Pöthig et al. 2010) is still a potential long-term P source for surface waters. Under the right conditions, this legacy P can be transported from soils to surface waters by surface

runoff and soil erosion, as well as by leaching from soils (e.g. Rowe et al. 2016, McCrackin et al. 2018).

**5 Conclusions**

The results of this study indicate that groundwater was the major source of stream water P especially during ecologically relevant low-flow conditions. Because of the limited quantitative P exchange between stream sediments and stream water, stream sediments that may have originated from agricultural soils eroded into the stream did not contribute significantly to

the SRP loads exported by the stream. Similarly we did not find evidence for inputs of SRP from shallow anoxic riparian wetlands. Rather, the seasonal variations in SRP concentrations in streams with summer maxima and winter minima can be explained by the varying contribution of groundwater to the overall discharge. Previous studies attributed the commonly observed high summer SRP concentrations (Ebeling et al., 2021) with the lack of dilution from wastewater point sources (Bowes et al. 2014) or the redox-driven mobilization of SRP from riparian wetlands (Dupas et al. 2017b). Here we found

evidence that also time-varying dilution of groundwater inflow due to changing proportion of groundwater to total stream discharge can also lead to this concentration pattern, and that the corresponding SRP concentrations during summer low flow can be well above the critical threshold for eutrophication. Deriving measures to reduce P concentration and fluxes in streams and rivers is therefore still a challenging task as top-down analyses of surface water quality may not clearly indicate P sources.

Our study has shown that different methods need to be combined to successfully identify relevant SRP flowpaths even in small headwaters. It is still uncertain how we can transfer the results from the Schäfertal to landscapes with other geologic, hydrologic, and land use characteristics, but this is a prerequisite for the implementation of effective measures for preventing eutrophication in agricultural streams and receiving water bodies.

**Acknowledgments**

We thank our colleagues at Helmholtz-Centre for Environmental Research (UFZ) for valuable discussions at the early stage of this study. We thank the UFZ Analytics Department (GEWANA) for performing the chemical analysis of the collected samples. We also thank the technicians at UFZ Magdeburg and UFZ Halle for their continuous efforts to maintain the monitoring activities at the Schäfertal. We thank Xiaoqiang Yang for support of figure preparation. We thank Gregor Ollesch for providing the long term SRP data. The monitoring equipment was partly funded by TERENO (www.tereno.net), financed by the German Federal Ministry of Education and Research (BMBF).

**Data availability**

Long term hydrological data sets have been published in Supplement of Yang et al. 2021. The long term SRP groundwater dataset used in this study is available in the supplement.

**Author contribution**

MR: conceptualization, methodology, formal analysis, investigation, writing- original draft preparation, visualization. JT: conceptualization, methodology, formal analysis, investigation, writing- original draft preparation. FR: investigation, writing-reviewing and editing, visualization. MS: investigation, writing- reviewing and editing, visualization. KK: methodology, investigation, reviewing and editing. BG: software, validation, visualization. FM: investigation, visualization, reviewing and editing. AM: conceptualization, methodology, formal analysis, investigation, writing- original draft preparation, visualization.

**Competing interests**

The authors declare that they have no conflict of interest.

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
