# Peer review of "Seasonal variation and release of soluble reactive phosphorus in an agricultural upland headwater in central Germany"

_Hydrology and Earth System Sciences, 2022_

## Author Comment (AC1)

**Revision notes of the manuscript**

Kindly note that in the revision notes, comments from the reviewers are marked with "**Comment**", while our responses are marked with "**Response**".
* * *
**Responses to the comments from reviewers:**

Reviewer 1

**Comment:** I enjoyed reading "Seasonal variation and release of soluble reactive phosphorus in an agricultural upland headwater in central Germany" by Rode et al. The authors investigated potential delivery flow paths for P during various baseflow conditions: groundwater (GW) discharge, hyporheic exchange through stream sediments, and soil drainage. This case study is valuable for understanding how P in agricultural catchments is hydrologically delivered to streams and buffered along the way.

I think the manuscript needs at least moderate revision to make the story more effective and clearer. Additionally, I am not so convinced that the GW P delivered to the stream is "geogenic" nor that sediment porewaters are not adding any SRP to the stream -- these points needs more nuance.

**Response:** We thank reviewer #1 for the positive comment. Regarding the concerns, please see our detailed point-to-point responses below.

**Comment:** Main criticisms
Inconsistencies in methods, results, and conclusions lead to discounting sediment porewaters:

**Comment:** The radon data are used to support conclusions about GW gains within the stream but much remains unclear. What were the radon concentrations in the GW wells? How was the degassing actually determined? Further, if stream turbulence is the key component behind degassing (L198), then why is the degassing of Rn so much greater in the very slow flow of the summer campaign compared to the winter one?
**Response:** The radon groundwater endmember that was determined from samples taken from three groundwater wells adjacent to the stream and two subsurface agricultural drains amounted to 23.2 kBq/m³. This value will be added to the text in sect. 3.3.
The way how the degassing was determined is comprehensively described in Schubert et al. (2020), sect. 2.2.2. In order to keep our manuscript as concise as possible, we prefer to not repeat this lengthy discussion/explanation but to rather cite the Schubert et al. paper. A related remark has been added to the text in sect. 2.2.1 of our manuscript.
Indeed, stream turbulence is a key component for degassing, as strong turbulence increases the concentration gradient at the water/air interface. Still, stream width and depth are quite influential as well, if the stream is wide and shallow. In particular, the width/depth ratio has major impact. A high ratio allows a strong gas exchange between water and air. As mentioned by the reviewer in another comment (L313) "the water level" was "very shallow for the September 2020 event". With a stream width of about 30 cm and a depth of only 2 - 3 cm, the large water/air interface per water volume below dominated the degassing intensity. In addition, the very low flow velocity of the stream water (around 0.07 m s-1) gave the radon time to escape. Even though turbulence played no major role in September, we cannot expect laminar flow of the water (due to the bed roughness of the stream). Thus, even though turbulence was not visible or obvious, the low water depth and the comparably large width of the stream resulted in an intense contact of the water with the water/air interface and hence in strong degassing. Therefore, (and as mentioned in sect. 3.3) the degassing of radon is much more intense in the summer campaign.

**Comment:** Given the issues noted here by the authors, it may help to consult the review by Raymond et al. 2012 to set an initial estimate of Rn degassing for this system based on basic stream hydrology information.

**Response:** As mentioned by the reviewer, Raymond et al. (2012) is a comprehensive review article. The paper discusses gas transfer velocities in streams and small rivers based on a metadata analysis of 563 gas tracer release experiments. On the other hand, the focus of the study that we discuss in our manuscript was not on gas transfer from surface water bodies. Hence we don't want to discuss the matter in a supererogatory extend and prefer to rather cite related publications. In the revised manuscript, we will mention Raymond et al. (2012) (sect. 2.2.1), since it discusses the issue of degassing comprehensively.

**Comment:** My impression of the Rn data is that flow in the summer campaigns was so slow that the gas transfer velocity for Rn was perhaps dominated by gas diffusion. The authors seem to corroborate this point on L313-315.

**Response:** Indeed, radon loss by gas diffusion played a significant role in the summer campaign. This influence was allowed for by the FINIFLUX model accordingly. The model itself is described in detail in Frei, S. and B.S. Gilfedder (2015).

**Comment:** Further, the water level must have indeed been very shallow for the September 2020 event (L313). For this event, using the reported velocity of ~0.07 m/s, discharge of 0.51 L/s, assuming the smallest stream width of 0.33 m (derived from 10 m / 30 as mentioned on L174-175), and further assuming a simple rectangular stream channel, I calculate a stream depth of 2.2 cm! (Relaxing the assumptions above may yield an even shallower depth.)

**Response:** The stream depth varied over distance, but about 2.5 cm gives a rough idea of the situation in September, indeed. We allowed for this rather challenging situation by adjusting the associated FINIFLUX parameters accordingly, reasonably and carefully. The FINIFLUX model itself is described comprehensively in Frei, S. and B.S. Gilfedder (2015).

**Comment:** Given that the stream was moving this slow and the depth was this shallow, I really doubt that sediment porewaters have such apparently little connection to SRP in the water column. Note that the sampled sediment porewaters 7 cm deep (L225) are relevant, but it's likely the uppermost 1 to 2 cm of the sediment that dominate P exchange between sediments and the water column. I suspect that SRP in the porewaters may possibly be lower towards the top of the benthos due to the oxygen gradient (see Palmer-Felgate et al. 2010 for example) but this still may not completely stop a significant flux of SRP to the overlying water column. It's been observed elsewhere that SRP can increase in summer months and that it's likely tied to redox status of benthic sediments (Smolders et al. 2017 cited in text; L75-79) -- why might this argument not apply in this case study?

**Response:** We found very high porewater concentrations in a depth of 7 cm in the downstream sampling point (20 times the concentrations in the stream water). We agree that the uppermost sediment layer will be dominant for exchange and likely have lower SRP concentrations. Here we do not have uppermost porewater concentration data proving this. However, the main evidence that this shallow stream does not take up significant amounts of P from the porewater is the observed water and SRP flux in the last 600 m of the stream. Data indicate that neither water nor SRP is gained here but that largest part of the flux comes from the upstream stream section. The downstream porewater show clear indication of reductive mobilization of iron and DP (manuscript line 355-356). Here we would make this more clear and add the Smolders et al. 2017 reference.

Note that the limited influence of shallow porewater in the upstream, strongly gaining, section is based on the different P speciation in porewater compared to the one observed in the stream water downstream of that section. Rather the gaining groundwater sampled upstream was similar to stream water. So we have three arguments here to underline the limited influence of the streambed SRP source: 1. Distinct water and SRP flux gaining in the upstream section but not in the downstream section. 2. Hydrochemical similarity of stream water and gaining groundwater. 3. Hydrochemical

dissimilarity of sediment pore water and stream water. We will improve our argumentation in section 3.4 to better pinpoint that.

**Comment:** Although it's possible that the more oxic sediments at the very surface should have a stronger sorption capacity than the deeper sediments, there's also the possibility of colloidal P -- generated from below -- bypassing the sorption sites and increasing the SRP signal (Gottselig et al. 2017). (I'm assuming filtered water samples [filter size not stated, 0.45 microns?], were still not small enough to prevent these colloids.) There seems to be enough Fe and DOC in the waters to support this.

**Response:** Our argumentation is not that the possible oxic boundary layer between sediment and water prevents P transport. We argue that the chemical composition (ratio SRP/DP, nitrate, ammonia) and the age of the DOC are very different in the pore water and in the stream. If there was efficient transport of solutes from the pore water to the sediment, then the chemical composition and age of the DOC in the stream would have to approximate the composition and age in the pore water - but this was not the case (lines 330-339, 346-350). It may well be that (colloidal-bound) P from below bypassed the sorption sites and was measured as SRP in the stream. In this case, however, it was not diffusion (from the boundary layer) but advective transport from deeper layers. This would have carried not only phosphate but also nitrate, ammonium and modern DOC into the stream and, if significant, would have approximated the chemical composition and age of the DOC. But there was no indication of this. We therefore assume that there was transport of P from the sediment by diffusion and advective transport through microchannels, but this was insignificant in extent. The pore size of filters was 0.22 μm. We will include this information.

**Comment:** It is inappropriate to treat SRP or DP as conservative endmembers mixing in this system, as text on e.g., L331-335 suggest is the basis for arguing that sediment porewaters are insignificant. A more compelling argument is needed. (Further, the data cited on these lines is from 7 cm deep so the DP there may be even less relevant.)

**Response:** We understand that SRP or DP alone cannot be considered conservative endmembers because, for example, the concentration of SRP can change by uptake in algae or bacteria or by precipitation with Fe. As stated above, however, we do not consider the concentration but the ratio of SRP to dissolved organic P (DOP) as being significant. We have referred to this as the fraction of SRP in dissolved P (DP), but would express this in the revised manuscript as the ratio SRP/DOP, where we define DOP as DP minus SRP. Thus, at the upper station, where the significant inputs of water and solutes occur, we would encounter an SRP/DOP ratio of 0.078 in the sediment and 0.935 in the stream. The assumption that turnover in the stream reverses the ratios of sediment-borne P would require dramatic conversion of DOP to SRP, which is unrealistic in such a short travel time in the small stream reach. We therefore believe that the SRP/DOP ratios are a good argument that supports our argumentation.

**Comment:** Only geogenic P in GW?
I find it surprising that GW P in this predominantly agricultural catchment (receiving regular P applications (L104)) is considered geogenic (paragraph starting L401). It seems that these soils have reasonably high P (L120-122). It is even stated on L125 that fluctuations in soil P "suggests a transport of soluble P compounds to deeper layers". [Note that this sentiment reverses later in the paper on L416-418.] So can we really assert here that there's no agricultural P reaching GW and, later on the flowpath, the stream?

**Response:** This is a good point. In the cited reference only the top soil layer was investigated, which means only the first top 2 cm. The main objective of the former study was to evaluate the transfer of soluble P from the top soil to surface runoff. Keeping the relatively high P concentrations in the top layer after fertilizer application in mind it is possible that soluble P compounds can be transferred to the underlying layers. On the other hand the loamy soils show a distinct reduction in TP and in $C_{org}$ – concentration with increasing soil depth. In a typical agricultural soil of the study catchment, TP concentrations were elevated in the Ap horizon (778 mg/kg) but showed a dramatic drop to a mean

value of 193 mg/kg in the B and C horizon below the plough pan in 40 cm depth, indicating a very high P sorption capacity of the soil. This indicate that P does not leach from the Ap horizon to deeper layers in the investigated soil (Kistner 2007). This is consistent with our data of tile drain runoff, which indicate lower DP concentrations than both surface water and groundwater. Because investigated tile drains only cover a small but wet area in the valley bottom of the study catchment other areas of the catchment may show different DP concentration in percolation water of the soils. On the other hand this is not very likely because high groundwater levels, as they are in the area where tile drains exist, may accelerate the exchange between soils and groundwater while long seepage distances in more remote areas from the valley bottom may increase sorption processes and increase DP retention (Wriedt et al. 2019). Although soil P leaching is not very likely in the study catchment we cannot completely rule out any loss from agricultural soils to groundwater. We will consider this in the revised manuscript. See also our response to the next comment.

**Comment:** It seems that true natural reference points for GW SRP (i.e. with no history of agriculture) would be needed to support this -- is this available in the Wriedt et al. 2019 reference?
**Response:** The study of Wriedt et al. (2019) reveal mean DP values in groundwater of hard rock areas of 0.03 mgPL$^{-1}$ with a range of 0.01 to 0.10 mgPL$^{-1}$. He did not find significant differences between land use types (arable land, grassland and forest) but higher outlier values occur more often under arable land compared to forest. In our study area mean DP groundwater concentrations where with 0.05 mgPL$^{-1}$ slightly above the mean background concentration and therefore agricultural impact cannot completely ruled out although our soil P and DP groundwater and tile drain data do not confirm DP leaching to deeper soil layers and into groundwater. We will consider this in the revision of the manuscript.

**Comment**: I take issue here because it is increasingly acknowledged that many of our agricultural catchments are dealing with P "legacies" which will take decades or longer to remediate even with drastic action. Wrongly attributing P sources can lead to even weaker action and longer times for surface waters to recover, if at all.
**Response:** We fully agree with this arguing. Although in Germany the P balance of agricultural land is closed since more than 20 years TP is mostly elevated in the top soil of agricultural land and often twice as high as natural background conditions (Schachtschabel et al. 1992, Pöthig et al. 2010). This legacy P is subject to transport via surface runoff and soil erosion and under certain conditions also via leaching through the soil and tile drains into surface waters (e.g. Rowe at al. 2016, McCrackin et al. 2018). Reduction measures of these legacy P loss to surface waters should therefore be targeted to the dominant transport pathways. As referenced in our introduction many catchments with certain soil and groundwater conditions may be susceptible for transport of legacy P also via subsurface flow. In our study we argue that under such specific catchment conditions with loamy soils and their high P absorption capacity soils P leaching may be less important compared to other transport pathways like surface runoff. We will consider the issue of legacy P in our discussion.

**Comment:** The discussion on L415-417 seems to ignore that N and P behave very differently in their transport; seeing faster movement of nitrate than for P is not surprising and doesn't support the statement on L416-417. In fact, the Dupas et al. 2017 paper cited in text argues this very point (and is summed up in their Figure 7).
**Response:** We agree with this comment because our reasoning is possibly not detailed enough. Indeed, compared to N leaching P leaching from soil is mostly low. This finding does not allow to relate this transport pathway to other P transport pathways. Multiple processes can change the availability of N and P in the groundwater and the further transport to the stream and N:P rations may change via the transport from soil to streams. Therefore the input from the soil is likely not a sufficient indicator for assessing the transport of these two compounds from the subsurface to the stream. Some misunderstanding is possibly caused by the fact that we did not clearly distinguished between soil processes and transport processes from soil to the stream. We will limit our rational to the leaching of DP from soil. Soil P profile data do not support significant transport of P from the Ap

soil horizon to deeper soil layers, thus SRP transport via seepage water can be assumed to be limited. We will consider this and revised the manuscript accordingly.

**Comment:** I actually concur with what's said on L426-428: SRP is probably well buffered in the catchment but that means that those sorption sites may slowly leak P to the system, maintaining ecologically relevant SRP concentrations in the stream for a long time.
**Response:** We agree with this comment. Although SRP leaching is very limited from the loamy soils in the study catchment we cannot completely rule out slowly leaking of P. Available data on SRP transport within the soil column do not suggest considerable SRP leaching from the soil but this data are limited to a single soil profile. Although the silt part of the soils dominate throughout the catchment hydromorphic properties may super-impose locally. Further long term studies on P leaching in soils are missing. Therefor possible long term leaking of P cannot be entirely ruled out and may slightly vary within the catchment due to varying soil properties.

**Other general comments:**
Stream description:
**Comment:** The catchment is described quite thoroughly, but what about the stream itself? Slope, width & depth at time of sampling, morphology type, substrate characteristics, light availability, etc. would be quite pertinent to this study.
**Response:** The stream itself has a length of 1747 m a slope of 2 %, with a mean discharge of 0.33 mm d$^{-1}$ (5.5 l s$^{-1}$). The stream has a mean width of 0.4 m and a depth of 0.05 m. The substrate consists of fine and mid-granular sand fraction. The light availability is very high, because of only a small forest area of 3 % of the catchment size, which stands mainly not along the stream.
**Comment:** Illustrating on Figure 1 or adding a new subplot with focus on the stream itself to show locations of all the measurements (tracers, sediments, etc.) would be great context.
**Response:** We will add a new figure including all measurement sites along the stream and its surrounding.

**Comment:** Suggest sticking to the red/blue for summer/winter throughout (e.g. avoid the change in Figure 7). Please try to fix the x-axes in the figures with stream distance (m) so that they're similar/more comparable across figures. The aspect ratio for Fig 6 is too wide. The changes in figure styles and dimensions when 'distance from upstream' is on the x-axis makes it more difficult to follow the story across figures.
**Response:** We will adjust the figures to be consistent in color schemes and the use of the x-axis.

**Comment:** Consider combining Fig 2 and 3 to save space. Figure 4 could also be combined with 2 and 3 by showing the observed groundwater level for the study period overlaid upon the prior 10 year average -- just an idea.
**Response:** We will combine Figure 2 and 3. Including also Fig. 4 with a fourth information level would make the figure to complex. Therefor we would like to leave Figure 4 as it is but we could put this figure in the supplement and refer to it in the text as follows: "The groundwater levels [m below surface] in the sampling period ranged between 0.5 and 1.1 m. Relating to the 3 sampling campaigns groundwater levels were at 0.65 m (Jan. 2019), 1.0 m (Sep. 2019) and 0.85 m (Sep. 2020) below surface."

**Comment:** Double y axes in general: My preference is to avoid them if possible. It clouds the comparisons between seasons and visual points of data do not immediately map to values on the y-axis (until I re-read the caption to know which is which)
**Response:** We can add the x-axes to all sub-plots and put (a)-(d) into the sub-plots.

**Comment:** In some cases, they're entirely unnecessary, such as in Figure 7. (Also, why not just plot both series on log10? Having double axes and different transformations is doubly confusing -- this applies to both radon figures.)
If double axes have to be used, I think a simple but very helpful improvement would be to color-code the y-axes text/title with the same red/blue color scheme for the points.
**Response:** We agree with the reviewer's preference of avoiding double y-axes. At the same time, we have to mention that her/his opinion is a rather subjective perception. We love double y-axes! Still, we color-coded the y-axes in Fig. 7, as suggested.

**Comment:** With all the data available in some of the figures, the summary stats in Tables 1 and 3 aren't necessary and can be removed to reduce clutter; just make references to the stats of interest within text.
**Response:** We agree with this and will revise the manuscript accordingly

**Comment:** I'm not sure the 3rd subplot in Figure 6 (with net change in SRP flux) is necessary -- it's simple enough to examine the pattern in the actual SRP flux data in the bottom subplot, in my opinion.
While the log-axes in other figures are more clearly denoted (e.g. Figure 8), it's not very obvious in Figure 9 -- making figure styles more consistent would help here.
**Response:** Fig. 6 agreed - we can take out 6c without a loss of information. We will adding ticks to 9a similar to Figs. 7 and 8 to make more clear that axes are logarithmic.

**Comment:** Writing and writing style:
I had forgotten all of Section 3 was results and discussion, as there wasn't too much discussion until much later. (The last few paragraphs felt like a sudden 'dump' of discussion.) Perhaps this paper is better suited to separate R & D sections?
**Response:** We disagree here. Yes, we state pure results in terms of concentration and fluxes. But later on (3.4 & 3.5) we do both together. Separating the results from discussion here would need a lot of repetition. However, we can make all this more clear in the chapter title: 1. Observed discharge and stream SRP concentrations. 2. Observed longitudinal water and SRP fluxes. With words like "assessing" and "integrating" in the last chapters we make pretty clear that we also discuss the results.

**Comment:** Please streamline the introduction to build its focus up to the study objectives. I think introducing the main sources in points (a) to (d) on L50-54 and elaborating each is a fair way to structure the intro, but it seems that structure was forgotten after the intro had covered point 'c' (sediments, L71-79); L80 onwards breaks with that structure leaving me wondering where the intro was headed. Further, if the intro is going to focus on "SRP mobilisation ... in various headwater compartments" (L50) as the first paragraph indicated, then lead sentences in paragraphs such as on L63 should more clearly follow that thread. I.e., 'SRP mobilisation' should be the common theme throughout and clearly connected.
**Response:** We will streamline the point d) accordingly and follow the suggestion on "SRP mobilization" and will revise the manuscript accordingly

**Comment:** Avoid excessive passive voice throughout. E.g., L336-337 could be rewritten to avoid the "were found" and so make the sentence clearer: "At the outlet, concentrations Fe, DOC, dissolved P, and NH4 were greater in the sediment pore water than in the stream (Table 2),..."
**Response:** good point, we will revise the manuscript accordingly

**Comments:** Proofreading for clearer English would improve the paper. E.g., I'd suggest revising the sentence on L273-274 to: "The spatial pattern of SRP flux largely followed the pattern observed for discharge." This keeps the emphasis on the main topic of this paragraph (SRP flux) rather than on

'spatial pattern of discharge' in the original sentence. Another example sentence that could be made clearer/more effective is L56-59.
**Response:** We will revise the manuscript accordingly

**Comments:** Please break up the paragraph on L401-439 (139 lines!). Further, consider incorporating the discussion with the broader literature done here more evenly throughout Section 3.
**Response:** We will consider this where appropriate

**Comments:** L177-178: I'm confused by this sentence. What is the 'time lapse' issue here and why should that matter if you're moving upstream?
**Response:** That was not the best term used here. We mean time interval of 30 min between injections and will change that accordingly.

**Comments:** Section 2.2.1: was there any potential for significant sub-daily variation in discharge or was it stable?
**Response:** During all the field campaigns the discharge was stable

**Comments:** L199-201: it is stated here that there are multiple ways to estimate the radon degassing from the stream -- which the authors described as a "crucial parameter" just above. So... how was this actually determined here in the present study?
**Response:** Schubert et al. (2020), sect. 2.2.2, discusses comprehensively how the degassing rate was determined. In order to keep our manuscript as concise as possible, we prefer not to repeat this lengthy discussion but rather to cite the Schubert et al. paper. A related remark will be added to the text of our manuscript in sect. 2.2.1.

**Comments:** Section 2.2.2: was there any reason for not measuring Rn in the September 2019 campaign? Also, why were sampling locations different from the salt tracer points?
**Response:** Redoing the Rn measurements was logistically not feasible in Sep 2019. Yes, the sampling locations were similar between both Rn measurements and tracer tests. We will make this more clear in the text.

**Comments:** L344-350: I don't have any expertise with radiocarbon dating but is it really the case that DOC in the stream is millennia old? Could there be more discussion with literature in this section? Additionally, there's no mention here about the stream metabolism involved. If there's enough light, much of the DOC is likely autochthonous for summer baseflow, especially considering the high nutrient availability.
**Response:** Yes, the DOC in streams can indeed be thousands of years old, especially if it is groundwater. This does not mean that the water is that old, but the old DOC may have been dissolved and mobilized by recent hydrolysis of old soil POC. In other streams in this area we have found DOC that had a radiocarbon age of up to 3000 years. We will discuss this briefly and include a reference.
The degradation of the DOC also depends on the light irradiation, but is largely determined by the residence time. In incubation experiments using stream DOC, low degradation rates between <0.5% and 2.3% per day were measured (Kamjunke et al. 2022). Thus, a significant change of 14C values by selective decomposition of young or old DOC is unlikely. We will discuss this briefly in the revised manuscript.

**Comments:** L354: Can more concrete evidence be given instead of stating "...probably transported by preferential flow paths."
**Response:** See also reply above. There is no additional evidence. The gaining groundwater sampled in the deep streambed has the same signature as the stream water further downstream. We do not see evidence that the shallow pore water (having a different signature) is significantly contributing to the stream flow chemistry. We therefore propose that the groundwater gains bypass the shallow

pore water. This is likely done by preferential flow paths. Schmidt et al. (2006) is a good example of this behavior showing by temperature-based flux measurement that a fine-grained streambed with low hydraulic conductivity favors preferential flow paths so that largest part of the incoming groundwater flux is channeled to a few distinct locations. We will revise this section and give this literature example as an additional argument.

**Comments:** L402: where in the paper are the 'oxidised groundwater conditions' established? Could this be included in Table 2?
**Response:** Right. This was not made clear before but can be seen in Tab 2. (arguing with the low Fe and NH4 concentrations). We will add this (Line 328 in the original manuscript) to chapter 3.4. We will also make a reference to Table 2 in this point of the manuscript.

**Comments:** L432: Please avoid the "source/sink" dichotomy for P and sediment sorption. It ignores the transient nature of sediment P and how, really, sorptive materials in catchments only buffer P.
**Response:** We will consider this in the revised manuscript

**Comments:** Technical comments
Please superscript all atomic mass numbers: e.g. 222Rn and 14C; fix subscripts too ("Psat" on L59, "NO3" on L69)
L30: 'SRP losses' here is kind of ambiguous in terms of the direction of the flux.
L31: 'SRP-fraction for dissolved P' is unclear.
L59: 'soil P sorption saturation' on its own doesn't mean greater P mobility unless you refer to soils with high P sorption saturation (typically, the saturation is expressed as some sort of degree). Additionally, is "Psat" ever used again in this manuscript? If not, no need for a new acronym.
L69: "NO3" should be defined.
L75: don't use "molybdate reactive P" if 'SRP' is used everywhere else
L89-90: suggest moving the list of pathways out of the parentheses as they're quite important
L90: "locate" instead of "localise"?
L92: the "gaining- and losing water fluxes" needs to be reworked, avoid the dangling hyphen if no hyphen is used for 'losing'
L112-114: this sentence doesn't seem to state anything clearly -- what's the message here?
L122: move the Kistner et al. 2013 reference up one sentence (to align with the "Previous research has..."). Also, no need to give a new acronym for DPS if that's not used again.
L150: I think there's a zero instead of "O" in the "NO3" here.
L150: The Dupas et al. 2017 reference here is out of place; save this for the discussion.
L213-215: Were the probes calibrated on the day of measurement? Please cite a reference for the 'standard methods' for the P analyses and give some note of accuracy and/or method detection limit.
**Response:** We will consider all technical notes in the revised manuscript

**Comments:** L217: how was dissolved iron measured? detection limit?
**Response:** Dissolved Fe concentrations were measured by ICP-OES (Perkin Elmer 7300 DV). The limit for determination was 0.01 mg L$^{-1}$.

**Comments:** L268: 'proportion' not 'proportionate'
L270: "neutral" is an odd term to use here, perhaps switch for "had little net change in discharge"
**Response:** We will consider all technical notes in the revised manuscript

**Comment:** L275: perhaps switch out 'gained' for 'contributed'?
Figure 7: please also add to the caption or indicate on the figure what the dashed line means (L286). And shouldn't this apply to Figure 8 too? Also, please keep the time labels consistent with other plots (i.e. January 2019 and September 2020 instead of 'winter' / 'summer').
**Response:** Actually, the water was "gained" by the stream (the groundwater "contributed" to the stream flow). The meaning of the dashed line is explained in the text above Fig. 7. Still, we added the

information to the figure caption, too. The information "January 2019" and "September 2020" has been added to the figure caption.

**Comment:** L310, how reliable are the values given here for groundwater given the text on L296? Is some value representing uncertainty (e.g. confidence interval) possible here?
**Response:** We state that the recorded data and the data evaluation based on the FINIFLUX model suggests that about 60 – 65 % of all water entering the stream is contributed by groundwater. The uncertainty of the result is mainly a function of the assumed radon degassing rate. Still, the related FINIFLUX parameters have been chosen carefully and are reasonable. A discussion of the FINIFLUX model including error propagation would be beyond the scope of the study. The model is described in detail in Frei, S. and B.S. Gilfedder (2015).

**Comment:** Section 3.4: edit the title to include "in summer baseflow (September 2020)" as that's crucial context in this whole section.
L336: make NH4+ more consistent throughout text (replace "NH4"); consider also including the valence for "NO3-"
**Response:** We will consider all technical notes in the revised manuscript

**Comment:** Table 2: why is the Fe in the outlet stream sample "n.d." (unsure whether this is "not determined" or "not detected")? Or was it below detection? If the below detection, what was the MDL?
**Response:** n.d. actually means not determined, but I have mixed this with a non-analyzed value from another sample. The correct value is 0.027 mg L-1 , which we enter in the table. Thanks for the hint.

**Comment:** Table 2: please add pH, dissolved oxygen (or some indication of redox status), conductivity, and temperature here as that would be very helpful context (and seems to have been measured according to 2.2.3).
**Response:** We will add these compounds within the table

**Comment:** L432: should this be 'House 2003'? This reference is missing
References -- suggest checking all:
Kleinman et al. 2009; van Dael 2020 reference is duplicated
also some refs are CAPITALIZED
LLFG 2021 reference (L104) is missing
**Response:** We will check and correct this references

**References cited**

Gottselig N, Amelung W, Kirchner JW, et al (2017) Elemental composition of natural nanoparticles and fine colloids in European forest stream waters and their role as phosphorus carriers. Global Biogeochem Cycles 31:1592–1607. https://doi.org/10.1002/2017GB005657

Palmer-Felgate EJ, Mortimer RJG, Krom MD, Jarvie HP (2010) Impact of point-source pollution on phosphorus and nitrogen cycling in stream-bed sediments. Environ Sci Technol 44:908–914. https://doi.org/10.1021/es902706r

Raymond PA, Zappa CJ, Butman D, et al (2012) Scaling the gas transfer velocity and hydraulic geometry in streams and small rivers. Limnology and Oceanography: Fluids and Environments 2:41–53.

Additional References:

Kistner, I., 2007: Anwendung des Modells ANIMO zur simualtion des gelösten Phosphors im Oberflächenabfluss auf der Feldskala und der Phosphorverfügbarkeit im Oberboden auf der Einzugsgebietsskala (in German). Dissertation, Martin Lutter University Halle,182 p.

Kamjunke N, Beckers, LM, Herzsprung P (2022) Lagrangian profiles opf riverine autotrophy, organic matter transformation, and micropollutants at extreme drought. Science of the Total Environment 828:154243. https://doi.org/10.1016/j.scitotenv.2022.154243

Scheffer, F. 1992: Lehrbuch der Bodenkunde/ Scheffer Schachtschabel.13. Auflage, Stuttgart: Enke Verlag, 491 p.

Pöthig, R., Behrendt, H., Opitz, D., Furrer, G., 2010. A universal method to assess thepotential of phosphorus loss from soil to aquatic ecosystems. Environ. Sci. Pollut.Res. 17, 497–504.

Schmidt, C., Bayer-Raich, M., Schirmer, M., (2006): Characterization of spatial heterogeneity of groundwater-stream water interactions using multiple depth streambed temperature measurements at the reach scale. Hydrol. Earth Syst. Sci. 10 (6), 849 - 859

---

## Author Comment (AC2)

**Revision notes of the manuscript**

**Kindly note that in the revision notes, comments from the reviewers are marked with "Comment", while our responses are marked with "Response".**
* * *
**Responses to the comments from reviewers:**

Reviewer 2

**Comments:** Main comments
This research addressed the question of SRP release to surface water during baseflow conditions in different seasons (Winter and Summer). The research combines measurements of SRP in stream water sections, in streambed sediments, sediment pore water and groundwater, as well as tracers' tests and Rn measurements to localize and quantify GW inflow. This is important to better understand P delivery and release to surface water and the respective role of the different P sources.

I think that it is a manuscript with great potential but with a lot of information that need to be more organised to have a clearer story. There are quite a lot of grammatical issues to correct throughout the manuscript, a thorough proofread is needed. There are some details in the material and methods section to correct/add (measurement methods, uncertainty…). The discussion comes also very late and is too short, without many references to other studies. I would separate the results and discussion sections and would develop the later by discussing YOUR results and the processes that may be involved.
**Response:** We thank reviewer 2 for the positive comment. Regarding the concerns, please see our detailed point-to-point responses below. We will consider a thorough proofreading.

**Comments:** Technical comments
I would improve the resolution of Figure 1 and the y axes (colours when the scales of the 2 y axes are different, labels).
**Response:** We will choose a higher resolution for Figure 1 and use different colours for the two y-axes in Figure 2

**Comment:** I would add one figure with all the sampling/measurements points, that would make things much clearer.
**Response:** We will prepare a catchment figure which includes all sampling points within the study site

**Comments:** Check the references to figures and tables (all need to be cited in text and be the right one).
**Response**: We will check all references and correct them accordingly in the revised manuscript

**Comment:** No abbreviations ("SRP") or numbers ("52%") at the beginning of a sentence.
**Response**: We will consider this in the revised manuscript.

**Comment:** Superscript and subscripts to check.
**Response**: We will consider this in the revised manuscript.

**Comment:** There is a lack of consistency in the terminology (Summer/Winter, September/January).
**Response**: We will consider this in the revised manuscript.

**Comments:** Specific comments

Line 20: I think there is a mistake in the email address.

Abstract:

Line 30: SRP-fraction for DP? It is unclear.

Introduction:

Line 50: I would try to improve the transition between the temporal variability of SRP concentrations and the different P sources, it is too quick as it is now.

Lines 56-59: I would suggest separating in two sentences, it is a long sentence that may be difficult to follow and understand.

Lines 63-66: I would also here separate the sentence in two.

Lines 68-69: The sentence about temperature-dependent processes and its link with the redox conditions discussed above are hard to understand, I would clarify this. The transition to temperature-dependant processes would need to be improved.

Lines 71-75: The first part of this paragraph is hard to follow and understand due to grammatical errors, lack of clarity and organisation.

"In situations, …": I would not use this, but would go directly to the point.

"fed by baseflows": I would say "fed by groundwater" or "during baseflow conditions".

"Data suggest…": which data? Reference needed here.

**Response**: We will consider all this technical notes all through the revised manuscript.

**Comments:** Line 79: I am missing here a paragraph about geogenic sources of P, as it has nicely been done for the other P sources in the above paragraphs.

**Response**: We will add a short new paragraph about geogenic sources of P.

**Comments:** Line 88: I am not convinced using the expression "headwater baseflows", maybe instead "in headwaters during baseflow conditions?".

**Response**: We will consider this suggestion accordingly.

**Comments:** Material & Methods

Lines 110 and 111: I would not use "e.g." when referring to conductivity values, I would give a range or a mean value instead.

Line 112: I think "circa" is commonly used before dates, so I would delete it and just keep "below 0.4 m".

**Response:** We will consider this through the revised manuscript.

**Comments:** Line 113: Which "detailed topographic characteristics"? I would give more details here.

**Response:** These topographic characteristics are slope and slope position, valley bottom and exposition. We will add these to the revised manuscript.

**Comments:** Line 121: DPS is Degree of P Saturation.

**Response**: This is correct. We will clarify this in the revised manuscript

**Comments:** Line 124: I would specify "declines of WSP in the topsoil…".

**Response**: We agree with this suggestion and consider this in the revised manuscript.

**Comments:** Lines 135: We go from precipitation/Q (Figure 2) to GWL (Figure 4) without mentioning air temperature (Figure 3). Either the data presented in Figure 3 should be presented in the text or Figure 3 should be deleted.

**Response**: We agree with the reviewer. We will combine Figure 2 and 3 and will refer to the new figure 2 in the text.

**Comments:** Lines 148 and 149: Change "between X-X" to "between X and X" as before.

**Response**: We agree with this suggestion and consider this in the revised manuscript.

**Comments:** Lines 146-150: To which period(s) (e.g. 2010-2020?) do these values refer to?

**Response**: Yes, the water age analysis of Yang et al. 2021 is relate to the period of 2010-2020 but the earlier water quality investigations are related to the years from 1999 to 2010. This is also true for the cited study of Dupas et al. (2017). We will add this information to the revised manuscript.

**Comments:** Line 152: I would write "January 2019 during a period of…" instead of "January 2019 with..".

**Response**: We do not see the need to change this sentence and would therefore keep it as it is.

**Comments:** Line 153-155: It reads like dilution tests and 222Rn measurements were also used to characterise stream water, groundwater and sediments, which is not true. I would correct this; the sentence can then be used to organise the section.
Here we have: 1) in-stream tracer dilution tests and Rn222 measurements to analyse lateral inflows and 2) what measurements? to characterise stream water, groundwater and sediments properties.

**Response**: We will follow this comments to better clarify the measurement and their purpose as follow: These campaigns comprised in-stream salt tracer dilution tests and 222Rn measurements in order to analyze lateral inflows to the stream, and water quality measurements to characterize stream water, riparian groundwater and stream sediment properties.

**Comments:** I would slightly change the headings of the section 2.2. to improve its organisation:
2.2.1. Lateral inflows to the stream
2.2.1.1. Water balance of stream sections measured by tracer dilution tests
2.2.1.2. Groundwater discharge investigated by Radon measurements
2.2.2. Stream water, groundwater and sediments chemistry

**Response**: This is an excellent suggestion and makes the purpose of the measurements much clearer. We will consider this suggestion in the revised manuscript.

**Comments:** Lines 169-191: This is a very good section; the methodology is clear.

**Response**: Many thanks for this comment.

**Comments:** Lines 203-204: How do these 6 locations relate to the 6 locations used for the tracer dilution tests? Are they different? If so, why? How far are they from each other?

**Response:** The Rn and tracer dilution test locations are similar. We will adjust the text and also refer to the new figure of the sampling locations here. The locations can be considered to be essentially identical.

**Comments:** Lines 208-210: I would develop on the method on how to get from Rn data to the localisation of groundwater discharge and its quantification. How is the rate of radon degassing determined?

**Response**: Schubert et al. (2020), sect. 2.2.2, discusses comprehensively how the degassing rate was determined at the Schäferbach. In order to keep our manuscript as concise as possible, we prefer not to repeat this lengthy discussion but rather to cite the Schubert et al. paper. A few explanatory details and a related remark will be added to the text of our manuscript in sect. 2.2.2.

**Comment:** Lines 212-228: I would improve the organisation of this section, maybe follow: field instrumentation-field sampling-field measurement-lab analysis? Which method did you use for iron analysis?

**Response**: We would like to keep the structure of the presented measurements because this is in line with the structure of the presentation of the complete set of field measurements which is organized according to the purpose and not the method.

**Response:** Dissolved Fe concentrations were measured by ICP-OES (Perkin Elmer 7300 DV). The limit for determination was 0.01 mg $L^{-1}$. We include this information in the revised manuscript.

**Comment**: Line 215: Please specify which methods you used (and with references) for P analysis, this is important. Include the method detection limit or uncertainty.
**Response**: Total (TP) and reactive phosphorus (SRP) were measured using the ammonium molybdate spectrometric method (DIN EN ISO 6878, 2004). Detection limit is 0.005 mgPL$^{-1}$. We will add this information to the manuscript.

**Comment**: Lines 229-235: This part of the section is more organised and easier to understand.
**Response**: Thanks for the positive comment.

**Comment**: Results & Discussion
Line 240: Change "in…" to "during the two summer campaigns…".
Lines 241-242: I would not use "highly" and "very".
Line 241: "Constant" instead of "consistent"?
Lines 241-243: This is a long sentence; I would separate it in two sentences.
Lines 243-244:  I would rather use a factor for comparison instead of a concentration.
Line 250: Should it be summer instead of autumn?
Line 256: I would specify in the heading "along the study reach of the stream".
Line 267: "proportion" not "proportionate".
Line 268: Do not start a sentence with number.
Line 270: The term "neutral" cannot be used here, I would rewrite the sentence.
**Response:** We agree with this technical suggestions and will consider them in the revised manuscript.

**Comments**: Lines 279-288: I would gather this part into a first paragraph presenting the results: longitudinal patterns and concentrations.
Lines 288-319: Then, in a second paragraph, I would discuss why we see these patterns and concentrations by bringing in the info on groundwater discharge localisation and rate. I think that would improve the organisation of this section.
**Response**: The reviewer might be right. However, we thought about the structure of sect. 3.3 quite intensely, too, and came up with the structure presented in the manuscript. First, we describe the recorded $^{222}$Rn patterns qualitatively, and then compare and discuss the quantitative differences of winter and summer result. Subsequently we describe the resulting groundwater discharge rates, again first qualitatively then quantitatively. We believe that this is the most appropriate way to describe the situation. Hence, we would like to stick to the structure of the section as it is.

**Comment:** Lines 296-309: More information in the methods section on the FINIFLUX model would help to better understand the uncertainties related to the modelled results.
**Response**: The FINIFLUX model is described in all detail in Frei, S. and B.S. Gilfedder (2015). The uncertainties of the model results, which are mainly correlated to the uncertainties of the applied degassing coefficients, are discussed in Schubert et al. (2020). In order to keep our manuscript as concise as possible, we prefer not to repeat this lengthy introduction into the model but rather to cite the Frei, S. paper and B.S. Gilfedder as well as the Schubert at al. paper for the more interested reader. Still, a few explanatory sentences on the uncertainty of the degassing coefficient have been added to the manuscript in sect. 3.3.

**Comment:** Line 312: I think you already said that before (lines 287-288?), in a different way. I would avoid repeating the results.
**Response**: We agree with the reviewer

**Comment:** Line 321: Why are you investigating sources of SRP only in September 2020? Explain why, it is not clear for me.
**Response:** -14C and pore water measurements were motivated by the finding of stream reach balances obtained in summer 2019. Furthermore we had financial restrictions for the use of the financially always costly 14C measurements. We therefore selected the most extreme and ecologically relevant period for SRP transport from land to water during end of summer only.

**Comment:** Line 322-324: It is hard to locate these observation points, a figure showing all sampling/measurement points would help a lot.
**Response:** We will consider this point and revise the Figure with sampling points "technical comments"

**Comment:** Line 356: "are at work", please rewrite this sentence.
**Response:** We agree with the reviewer and revise the sentence in the revised manuscript.

**Comment:** Lines 351-364: There are some good things here but there is no references at all to support your points, and the discussion about the underlying processes is almost absent. The discussion needs to be developed.
**Response:** We will modify the sentence in line 355-356 to read and add: „Sediment pore water, … exhibited high concentrations of DP, Fe and NH4+ but low concentrations of NO3- …, suggesting that reductive conditions predominated. This indicates that sediment was a source of P to the stream, but that transport was quantitatively limited and probably dominated by diffusive processes." We further add more discussion and two references to line 360: „The young radiocarbon age of the stream DOC suggests that its source lies in shallow sediments rich in organic material such as riparian soils. In contrast, the presence of aged DOC in groundwater, here with a radiocarbon age of more than 2000 years is typical of low flow periods (Schiff et al. 1997, Tittel et al. 2022). The groundwater itself need not be that old, the DOC may have been dissolved and mobilized recently by hydrolysis of old organic carbon from the soil."

**Comment:** Lines 374-386: It is too much focused on presenting the data, and not enough on discussing them. I would discuss briefly how your results compare with long-term data ("our results are consistent with...") but discuss more about YOUR data and the processes explaining what you observed.
Line 374-375: I would rewrite this sentence, some grammatical issues there.
Line 385: Any reference to support the suggested dilution pattern?
**Response:** We see the point raised here and will add more linkages to the seasonal campaigns. The dilution pattern is an observation from the data here. We would add more discussion and also references to comparable dilution patterns in Germany and elsewhere after this section as the lines 374-386 are a description of results and not a discussion which follows from line 394.

**Comment:** Line 394: I feel like the REAL discussion starts here, so very late. I think separating the results and discussion would be beneficial.
**Response:** We disagree here. Yes, we state pure results in terms of concentration and fluxes. But later on (3.4 & 3.5) we do both together. Separating the results from discussion here would need a lot of repetition. However, we can make all this more clear in the chapter title: 1. Observed discharge and stream SRP concentrations. 2. Observed longitudinal water and SRP fluxes. With words like "assessing" and "integrating" in the last chapters we make pretty clear that we also discuss the results.

**Comment**: Line 414-426: I really like this part where you discuss your findings, related them to land use and soil type. Some references are missing when you refer to other studies in the same catchment.
**Response**: Many thanks for this positive comment and we will add related references to the discussion in the revised manuscript

**Comment**: Line 418: Please rewrite the beginning of the sentence, it does not seem right.
**Response**: We will change the beginning of the sentence to make the context more clearer

**Comment**: Figures/Tables:
Figure 1: This figure is hard to read, the resolution needs to be improved. In the legend, I would not use "soil types" since it does not refer to WRB soil types. Maybe hillslope or topographical position?
**Response**: We will revise the figure accordingly as already considered in the response of Reviewer 1.

**Comments**: Figures 2/3/4: I would gather the three figures and use panels. Units of the y axes should be in square brackets.
**Response**: We would like to show only a combination of figure 2 and 3 and put Figure 4 in the supplement. Regarding Figure 4 please see the next response.

**Comments**: Figure 4: I would show only the same period as in Figures 2 and 3 in the text, a longer time series can be shown in the Supplement maybe. Is there an issue with where the vertical lines are located? It does not look to be the same dates as in Figures 2 and 3.
**Response**: There was a mistake in the timing in Figure 4. We will correct that and put Figure 4 in the supplement but referring to the groundwater levels of the campaigns in the text as follows: "The groundwater levels [m below surface] in the sampling period ranged between 0.5 and 1.1 m. Relating to the 3 sampling campaigns groundwater levels were at 0.65 m (Jan. 2019), 1.0 m (Sep. 2019) and 0.85 m (Sep. 2020) below surface."

**Comments**: Figure 5: Maybe consider colouring the y axes (blue/red) so it is easier to see that they have different scales.
**Response**: We will adjust the color of the axes as suggested.

**Comments**: Figure 6: In the caption I would use "during the three sampling campaigns" and not "in". Should it be "SRP net flux" instead of "SRP net" in the y axis title? Maybe consider colouring the y axes (blue/red) so it is easier to see that they have different scales between January and September. I would also change the order of the panels since Q net is calculated from Q and SRP net flux is calculated from SRP flux: Q, Q net, SRP flux, SRP net flux.
**Response**: We agree and will adjust the figure captions, order and axes. See also our response to reviewer 1 on the X-axis.

**Comments**: Figure 7: I would add a sentence explaining the dashed line in the caption (even though it is already in the text), so the reader does not have to look for it in the text. In the y axes titles, you use here winter/summer but in Figures 5 and 6 you use January/September, I would stick to one of them and not mix the two, be consistent. I would also change the colours to red and blue to be consistent with the other figures and I would colour the y axes (blue/red).
**Response**: We color-code the y-axes in Fig. 7, as suggested. Furthermore, the information "January 2019" and "September 2020" will be added to the figure captions of Figure 7 and 8. Even though the meaning of the dashed line (Fig. 7) is explained in the text above Fig. 7, we add the information to the figure captions of Figure 7 and 8, too, as suggested by the reviewer.

**Comment**: Figure 8: What is the uncertainty of the modelled Rn concentrations?
**Response**: The uncertainly of the result is mainly a function of the assumed radon degassing rate. The related FINIFLUX parameters have been chosen carefully and are reasonable. Still, a detailed

discussion of the FINIFLUX model (including error propagation related to the degassing coefficient) would be beyond the scope of the study presented here. Tor the more interested reader the FINIFLUX model is described in detail in Frei, S. and B.S. Gilfedder (2015). Uncertainty and error propagation related to the FINIFLUX degassing coefficient is discussed in Schubert at al. (2020). Still, a few explanatory sentences on the uncertainty of the degassing coefficient will be added to the manuscript in sect. 3.3.

**Comment:** Figure 9 : Why are there only 3 points here? And not all the measured concentrations points? Is it the average of each campaign? I am surely missing something here, I would clarify.
**Response**: Because all historical sampling data have been carried out at the outlet of the headwater catchment in Figure 9 we used only the samplings at the catchment outlet of the three campaigns. Due to nutrient processing within the stream reach we did not use upstream nutrient measurements from our campaigns for this comparison.

**Comment**: Table 2: "nd" refers to "not determined" or "not detected"? If it refers to not detected I would say "< MDL" instead. These MDL need to be given in the method section.
**Response**: n.d. actually means not determined, but I have mixed this with a non-analyzed value from another sample. The correct value is 0.027 mg L-1 , which we enter in the table. Thanks for the hint.

**Comment:** Table 3: Where do these data come from? Any reference?
**Response**: The origin of the data is described in the text above. Surface and drainage water samples are taken from Ollesch (2008) while groundwater data came from an internal campaign. We can add this information to the caption.

**New References**

Schiff SL, Aravena R, Trumbore SE, Hinton MJ, Elgood R, Dillon PJ (1997) Export of DOC from forested catchments on the Precambrian Shield of Central Ontario: Clues from 13C and 14C. Biogeochemistry 36:43–65.

Tittel, J., Musolff, A., Rinke, K., Büttner, O. 2022. Anthropogenic transformation disconnects a lowland river from contemporary carbon stores in its catchment. Ecosystems 25: 618-632.

---

## Referee Report (RR1)

Review for "Seasonal variation and release of soluble reactive phosphorus in an agricultural upland headwater in central Germany" by Rode et al. (hess-2022-126; submitted to HESS 2022-4-8; major revision 2022-9-1)

**General comments**

This is Referee #1 again, apologies for my slow review. I think the manuscript is still in need of some major revisions before consideration for publication.

I previously pointed out the inaccuracy and danger in (over)interpreting these data to conclude that P in this stream is "geogenic" – I disagree on this point even more now – and the authors seemed to concur somewhat in their response… yet the abstract and discussion remains unchanged on this point. The ultimate source of P measured in the stream is only, at most, a secondary point of interest for this paper: the actual objective was on differentiating between proximal P inputs to the stream (riparian wetlands, stream sediments, and deeper groundwater) during low flow. This study was not designed to investigate whether "intensive arable land use within the catchment was the cause" [L453] – the frame of reference in this study is much further down the flowpath, closer to the stream. Further, I'll discuss below why the discussion on the source of P in deeper groundwater for this catchment is highly uncertain. So, I insist this argument for geogenic P as the primary P source in the stream be removed entirely since (1) it's unsubstantiated, (2) it's not the point of the study, and (3) it's ultimately harmful for the management of P in catchments, very much like the one in this study, that face P legacies.

Related to the above comment, the introduction gives the impression that "land-to-water" SRP fluxes (L103) and a connection to land use (L93-96) is a focus of the study. But this study was not designed to address this. The methods used here cannot address this land use question because they cannot differentiate between ultimate P sources once they've already reached an intermediate zone – the "potential source zones" on L237 – such as groundwater. So the text should be focused to the actual parameters of the study.

Another major concern is the lack of a cohesive "story" in the paper. For example, the Conclusions is not coordinated with other sections in the paper (particularly the Abstract), doesn't really follow from the discussion, and has this sudden appearance of soil erosion as a major topic (why is this just now showing up?). The results + discussion (Section 3) doesn't have clear points anchoring it (see also my comment advocating for separate results and discussion). The mention of "riparian wetlands" in the abstract and introduction gets virtually no follow-up for the rest of the paper. Et cetera. Overall, the paper needs much more global cohesion to be effective.

Relatedly, I strongly suggest separating the results and discussion sections as it suits this particular paper much better. I think Reviewer #2 also pointed this out. For most of Section 3, the reader is waiting for the story of the paper to come together (since we're told this is also "Discussion"), which really only starts to take form in the latter half of 3.4 and then 3.5. Since sections 3.1 – 3.3 are mostly results, 3.4 is half/half, and 3.5 is all discussion anyways, it is not a stretch to simply condense results into its own section (along with some editing for conciseness), then have a well-rounded Discussion that pieces together all the evidence to tell the story. This edit would not require "a lot of repetition" of results. In fact I think the text would either be the same length or even shorter. The story of the paper might then be clearer and more effective.

The paper has a major focus on groundwater (GW) being a dominant source of P to the stream (e.g. paragraph on L440) and at times the text seems to suggest it's virtually the only source of P or the only process that matters (e.g. the text excludes/discounts other potential sources but doesn't provide/support additional, potential sources). However, just based on simple mass balance, this doesn't add up. Taking a GW discharge value of 0.75 L/s (Fig 6) and a GW SRP value of ~40 to 50 ug P/L, that gives a GW P load of 0.03 to 0.04 mg P/s. But this is considerably lower than the winter

total stream SRP load (~0.055 mg P/s) yet considerably greater than both the summer total stream SRP loads (~0.02 mg P/s). I think the authors should address this and discuss reasons for the disparity for both winter and summer conditions. In my view: other sources were mobilised in the winter event and in summer there may be in-stream/hyporheic processes that diminish/buffer the GW P flux.

**Specific comments**

Again, language throughout could be greatly improved. I've pointed out only some examples (not exhaustive) below in technical comments. This applies not only to grammar but also to conceptual and technical language.

I also have several specific comments below around technical aspects of the study which I think need refinement; these edits should also greatly aid the story of the paper.

A key point throughout this study is the connection of (deeper) groundwater to the stream under low flow conditions, supplying virtually all the flow, which coincides with generally the highest SRP concentrations. While I agree on this water source being prevalent at low flows, I'm not sure this study establishes this as the sole source for any given low flow period, as is illustrated with the two September campaigns. We only see detailed data for September 2020, which seems demonstrative of the groundwater P hypothesis, but notably September 2019 had much greater stream SRP concentrations (up to ~60-70 ppb). Unless deeper groundwater also increased SRP by 20-30 ppb, which I wouldn't expect as usually groundwater P is less dynamic, this high stream SRP leaves us wondering what the story is for September 2019 since groundwater SRP is likely lower. (Is there groundwater SRP data for this campaign too? I'm not seeing it.)

Further on the groundwater SRP: these groundwater SRP values are really high in general. I've read through the Wriedt et al. 2019 reference (mein Deutsch ist schlecht aber ich weiß genug) and I'd assume that the Schäfertal catchment would classify as part of their "Bergregion" (being part of the Harz mountains/Berge) – the Bergregion generally sees groundwater "phosphate" [note they analysed P via molybdenum-blue, i.e., it's SRP] concentrations of roughly 50 ppb as $PO_4$ or less (hence, around 16 ppb or less as P) depending on depth, etc.. The Schäfertal catchment in this current study, however, seems to have much greater groundwater SRP concentrations than this natural reference of 16 ppb SRP (despite its oxidated status), with Figure 7 suggesting a median around 50 ppb P (more than 5-fold greater than the natural reference). Having worked in catchments with comparable geologies (greywacke), I would be floored by 50 ppb SRP being "natural" or "geogenic". This is all to say: I sincerely doubt the dominant source of P in the study stream as being "geogenic". I'm not suggesting that the text explores this point – as it's not the point of the study – but rather that the authors reckon with this and remove unnecessary/unsupported text from the Discussion, then refocus the text on how the *proximal* sources of P coupled with hydrological flowpaths determines in-stream SRP for their three sampling campaigns.

Relatedly, the authors argue for the catchment soils being highly P sorptive and thus not conducive to P leaching to groundwater (L467-468). A change in soil organic matter content and total P with depth is not necessarily an argument for limited soil P leaching (L467) since P sorption capacity is not a function of total P and is only somewhat related to organic matter. We'd need more direct observations of P sorption capacity, such as the degree of P saturation (DPS). In fact, the Kistner et al. (2013) in-text reference gives topsoil (0-5 cm) DPS in the Schäfertal of roughly 31% -- this is fairly high and saturated to the point that the soil is conducive to P loss in either surface or subsurface runoff (Fischer et al. 2017; Kleinman 2017). Further P loss in the subsurface soil would depend more on the soil's sorption chemistry, which is unavailable. So I am not convinced that soil P (particularly that coming from the historical fertiliser use in this predominantly agricultural catchment) here isn't reaching groundwater.

While it's ok (in my humble opinion) to use the term "*soluble* reactive P", it does seem silly to use that term but then call the unreactive P in the same filtered water sample as "*dissolved* organic P" (I'm assuming that's what "DOP" stands for, as it wasn't defined). Additionally, I'd be cautious about calling the unreactive P in a filtered sample "organic" – it likely is predominantly organic P but won't necessarily be 100% organic P (simply "unreactive", as in "molybdate-unreactive", is more accurate).

L234: Dissolved P (DP) was not defined in the methods (L266). Is this total dissolved P? If so, I suggest including "total" in the name (i.e. TDP).

A focus of the paper is on baseflow conditions and the winter (January 2019) campaign is referred to as baseflow (L312). I think this needs to be more careful, as I'm not sure this event is really 'baseflow'. Discharge during the January campaign looks to me like a receding limb following a rain on snow event (Fig 2) – not exactly normal baseflow. Groundwater contribution as % of total discharge would be somewhat lower then (groundwater meaning what your groundwater endmember represents). Instead, much more Q is likely coming from runoff and shallower stores (soil matrix). [In contrast, the summer September campaign is thoroughly deep groundwater, which the model results also support. This makes sense to me hydrologically and strikes me more as 'baseflow' as I understand it.] Perhaps the authors just need to clarify what is meant by 'baseflow'.

L301-302: So, the 'groundwater' endmember includes not only groundwater wells but also agricultural drains, and apparently these water sources have fairly similar Rn concentrations. This means it's difficult, based on the Rn data, to differentiate between these water sources to the stream. Is it possible at some points that the drains are delivering P to the stream? At very low baseflow, couldn't this P be remobilised into the water column? At the very least, it's confusing how the different subsurface sources of water sometimes get lumped into "groundwater".

L308-312: This section should be deleted. First, it doesn't make sense to compare the Rn *concentrations* across different conditions anyways. The Rn tracer will degas back to atmosphere very differently across conditions. The summer low flow likely allowed tons of Rn degassing despite how much the GW contributes to Q. This Rn concentration comparison just has no place here. Second, the text here seems to imply that discharge in January was mostly GW while in September it was less so – the rest of the text argues the opposite. Absolute GW discharge could've been higher in January (although the data/model results suggest the GW Q's were pretty similar, which seems about right to me), but it made up proportionally *less* of the total Q.

L318-327: The sensitivity analysis here is nice to have. The text here demonstrates why I previously recommended giving an *a priori* value for the degassing coefficient based on the Raymond et al. 2012 review (see their empirical equations). I think this would bolster the discussion on this point. This is important because, for example, the degassing may be much greater than that parameterized for the winter event which implies that there must be greater 'groundwater' input (including the agricultural drains) in the first ~450 m which is also where the greatest increase in SRP flux occurs. I suspect the degassing may be too low because of the poor model fit in the downstream sections. Additionally, could a sensitivity analysis be performed for the summer campaign too?

Section 3.4: SRP:DOP means little, has no biogeochemical basis, and doesn't suit as a tracer; just focus on the SRP and DOP masses themselves. You can make the same point but with much more clarity by just focusing on DOP concentration – the ratio is just a distraction.

Section 3.4: This section repeatedly refers to porewaters from 7 cm deep as "shallow". Considering that stream itself is only 1 to 3 cm deep and that the substrate is fairly fine (L130), 7 cm below the substrate is actually very deep in the hyporheic zone, if it can be considered hyporheic at all (and remember: sediment porewater's connection to the water column is mostly via hyporheic exchange). Contribution to total hyporheic exchange tails off with depth enormously so it's likely the waters down here move extremely slowly and represent a tiny minority of total hyporheic exchange flow.

Boano et al. 2014 discusses this in detail. Any discussion of sediment should make it clear this is relatively \*deep\* sediment, and so sentences like L390-391 should be given better context. [Even with the reductive dissolution of Fe oxide bound P, it isn't clear what effect this has on the water column because this porewater must travel upwards through the hyporheic zone and likely comes across the zone of DO penetration and therefore potential for re-sorption.]

**Technical comments**

[Again, these aren't exhaustive. Please proofread carefully throughout.]

L23: "…*often* with the highest concentrations"

L27: "*of* SRP fluxes"

L28: revise to "… and sampled for SRP, iron, and $^{14}$C-DOC…"

L33: "and was thus…" – this logical connection doesn't make sense here because the abstract doesn't establish for the reader key points such as (1) SRP is high in groundwater and (2) summer low flow is dominated by GW (deeper GW, rather than other shallower sources). Please improve the connection.

L35-36: Remove "Examination of"; replace "confirms" with a verb more along the lines of "corroborates" or similar – temper the tone with scientific caution.

L37: I'm not sure what's being argued here with the bit on 'seepage from agricultural phosphorous'. [Note that P doesn't end with "ous"!] What does this mean to the reader at this point in the abstract?

L53: Stream sediments don't just provide P via reductive dissolution, there's also desorption (among other processes).

L56: I would advise against strongly scientific certain language like "undoubtedly". Could just revise this sentence to say "Subsurface transport is often dominated by preferential flow…"

L58: "therefore"

L59: How does a factor "such as soil P sorption saturation ($P_{sat}$)" "greatly increase P mobility through soils" ? Do you mean, soils with a *high* P saturation?

L69: Nitrate is an anion: $NO_3^-$

L74: I don't think "monomeric" is the term you mean here (I don't think phosphate counts as a monomer…). Do you mean "orthophosphate", i.e., phosphate in its various hydrated states ($HPO_4^{2-}$, $H_2PO_4^{1-}$, etc.)?

L75: "Molybdate reactive P redox-mediated release" – this isn't clear as written; please proofread throughout. Also, I pointed out in the previous review that it does not make sense to use 'molybdate reactive P' in one section but 'soluble reactive P' or other variants elsewhere. The 'reactive' implies 'molybdate reactive'. It's fine to me to use 'molybdate reactive P' in the manuscript but if so, use it throughout to avoid confusion. (The literature is full of varying definitions of the same analyte; let's not muddy the waters more.)

L93: Rather than 'level', just use 'concentration' for consistency.

L129: 1747 meters?

L130: Delete 'fraction'

L131: Revise the forest area part for grammar.

L132: TP is not defined.

L136: "DSP" – was this supposed to be 'DPS'?

L158: "runoff" seems like the wrong term here; consider "…under these drought conditions, water from deeper, older storage contributed more to stream discharge, particularly after a wet season [or in comparison to a wet season?]."

L163: Edit to: "Baseflow data show clear seasonal variations, with $NO_3$ peaking in winter while DOC and SRP peak in summer." [is the Dupas et al. 2017 reference specific to your study stream? Does it contain this data? Unless the answer to both is 'yes', then remove the reference]

Should the title of section 2.2.1 be broader than just "lateral inflows"? E.g. 2.2.1.1. doesn't really cover "lateral" inflows, but stream discharge itself. Maybe "Stream hydrological measurements" in general, or similar.

L212-213: This sentence adds little here. Instead, could you add here or perhaps in the results what gas transfer velocity could be expected (independent of your model) for your study stream based on the relationships provided by Raymond et al. 2012? I.e. give a value or range of plausible values. This would provide confidence in the parameter value obtained via your FINIFLUX model calibration.

Section 2.2.1.2. This section needs to be more clearly organized and probably should feature some paragraph breaks. The point raised on L209 ("A crucial parameter… is the rate of radon degassing…") just gets brought up but is left unresolved.

L217: "traces" – "tracer"

L226: Item (iv) is unclear to me.

L235: "*soluble* reactive P (SRP)"

L239: switching between "Fe" and "iron"…

L244: remove the comma after "both"

L253: italicize the "*g*" for the gravitational constant

Figure 5: The axis on the right side has a "0.0" label – this is inconsistent with the log scale. Edit this axis to be more like that in Figure 6B.

L339-340: This sentence confused me because I knew the January 2019 total discharge was much greater (5 to 6 L/s) until I realised this sentence referred to the water gained *within* the study reach – perhaps edit to make this clearer.

L341: 'stream depth' instead of 'water level'

L342: It's not necessarily that there was "intense radon degassing" during this low flow – degassing would be much more intense due to turbulence (a key influence of radon loss) during higher flow, no? I think it's just that stream velocity was very slow so the time available for degassing was much greater.

L356: Why isn't electric conductivity included in Table 1? This is actually the closest thing to a 'conservative tracer' and can help compare the different waters.

Table 1: It's not clear what the distinction is between the two groundwater entries are (perhaps explain in caption?). Also, maybe it's just the formatting, but should the second GW entry have "n=6" as the station?

L388: Before this point (e.g. L232), I was already wondering why nitrate wasn't in Table 1 but sentences like this especially highlight this – can nitrate be added?

Figure 7: Clarify the sources for the historical data here (they're not from this study, correct?) in the caption.

**Additional references**

Boano, Fulvio, Judson W Harvey, A. Marion, Aaron I Packman, R. Revelli, L. Ridolfi, and A. Wörman. "Hyporheic Flow and Transport Processes: Mechanisms, Models, and Biogeochemical Implications." *Reviews of Geophysics* 52 (2014): 603–79. https://doi.org/10.1002/2012RG000417.Received.

Fischer, P., R. Pöthig, and M. Venohr. "The Degree of Phosphorus Saturation of Agricultural Soils in Germany: Current and Future Risk of Diffuse P Loss and Implications for Soil P Management in Europe." *Science of The Total Environment* 599–600 (December 1, 2017): 1130–39. https://doi.org/10.1016/j.scitotenv.2017.03.143.

Kleinman, Peter J. A. "The Persistent Environmental Relevance of Soil Phosphorus Sorption Saturation." *Current Pollution Reports* 3, no. 2 (June 1, 2017): 141–50. https://doi.org/10.1007/s40726-017-0058-4.

---

## Author Response (AR2)

**Revision notes of the manuscript**

Kindly note that in the revision notes, comments from the reviewers are marked with "**Comment**", while our responses are marked with "**Response**".
* * *
**Responses to the comments from reviewers:**

**Reviewer 1**

**Comment:** This is Referee #1 again, apologies for my slow review. I think the manuscript is still in need of some major revisions before consideration for publication. I previously pointed out the inaccuracy and danger in (over)interpreting these data to conclude that P in this stream is "geogenic" – I disagree on this point even more now – and the authors seemed to concur somewhat in their response… yet the abstract and discussion remains unchanged on this point.
The ultimate source of P measured in the stream is only, at most, a secondary point of interest for this paper: the actual objective was on differentiating between proximal P inputs to the stream (riparian wetlands, stream sediments, and deeper groundwater) during low flow. This study was not designed to investigate whether "intensive arable land use within the catchment was the cause" [L453] – the frame of reference in this study is much further down the flowpath, closer to the stream. Further, I'll discuss below why the discussion on the source of P in deeper groundwater for this catchment is highly uncertain. So, I insist this argument for geogenic P as the primary P source in the stream be removed entirely since (1) it's unsubstantiated, (2) it's not the point of the study, and (3) it's ultimately harmful for the management of P in catchments, very much like the one in this study, that face P legacies.

**Response:** We thank the reviewer for the comment and agree that the major aim of this paper is to elucidate proximal input pathways of SRP in the stream. In the study we can show that groundwater inputs are the dominant source of instream SRP at low flow conditions. As indicated in our former response we agree that specifying the sources of SRP in the groundwater is still associated with uncertainties. We therefore made changes in the abstract (see also below). Although the identification of sources of SRP in the groundwater was not the main objective of this study, we should provide the reader a brief explanation/discussion from which sources the SRP within groundwater may stem but, at the same time, acknowledge the uncertainties. We refer here to results from earlier studies. Therefore we would like to keep the discussion on sources of SRP in the groundwater. Nevertheless we discuss this issue more carefully and prevent stipulations in the discussion section.  See also our comments below.

**Comment:** Related to the above comment, the introduction gives the impression that "land-to-water" SRP fluxes
(L103) and a connection to land use (L93-96) is a focus of the study. But this study was not designed to address this. The methods used here cannot address this land use question because they cannot differentiate between ultimate P sources once they've already reached an intermediate zone – the "potential source zones" on L237 – such as groundwater. So the text should be focused to the actual parameters of the study.

**Response:** We agree with the referee to make the focus of the manuscript more clear and modified Line 103 and used "proximal SRP" instead of "land-to-water SRP". Further we specified the line 93-96

more clearly "(transfer from deep groundwater,…)" and "to localise the major source areas of SRP along the stream".

**Comment:** Another major concern is the lack of a cohesive "story" in the paper. For example, the Conclusions is not coordinated with other sections in the paper (particularly the Abstract), doesn't really follow from the discussion, and has this sudden appearance of soil erosion as a major topic (why is this just now showing up?). The results + discussion (Section 3) doesn't have clear points anchoring it (see also my comment advocating for separate results and discussion). The mention of "riparian wetlands" in the abstract and introduction gets virtually no follow-up for the rest of the paper. Et cetera. Overall, the paper needs much more global cohesion to be effective.

**Response**: Thanks for this comment and we revised the manuscript accordingly by separating results and discussion and considering the riparian wetlands in all manuscript sections appropriately.

**Comment:** Relatedly, I strongly suggest separating the results and discussion sections as it suits this particular paper much better. I think Reviewer #2 also pointed this out. For most of Section 3, the reader is waiting for the story of the paper to come together (since we're told this is also "Discussion"), which really only starts to take form in the latter half of 3.4 and then 3.5. Since sections 3.1 – 3.3 are mostly results, 3.4 is half/half, and 3.5 is all discussion anyways, it is not a stretch to simply condense results into its own section (along with some editing for conciseness), then have a well-rounded Discussion that pieces together all the evidence to tell the story. This edit would not require "a lot of repetition" of results. In fact I think the text would either be the same length or even shorter. The story of the paper might then be clearer and more effective.

**Response:** We agree to consider the riparian wetlands in more detail in the result and discussion section. Moreover we now separated the result and discussion section, see the revised manuscript.

**Comment:** The paper has a major focus on groundwater (GW) being a dominant source of P to the stream (e.g. paragraph on L440) and at times the text seems to suggest it's virtually the only source of P or the only process that matters (e.g. the text excludes/discounts other potential sources but doesn't provide/support additional, potential sources). However, just based on simple mass balance, this doesn't add up. Taking a GW discharge value of 0.75 L/s (Fig 6) and a GW SRP value of ~40 to 50 ug/L, what gives a GW P load of 0.03 to 0.04 mg P/s. But this is considerably lower than the winter total stream SRP load (~0.055 mg P/s) yet considerably greater than both the summer total stream SRP loads (~0.02 mg P/s). I think the authors should address this and discuss reasons for the disparity for both winter and summer conditions. In my view: other sources were mobilised in the winter event and in summer there may be in-stream/hyporheic processes that diminish/buffer the GW P flux.

**Response:** We do not argue that groundwater is the only SRP source in the Schäfertal catchment but the major SRP source during low flow. Groundwater SRP flux in the winter campaign can be approximated using the observed groundwater concentrations and the Radon-based inflow of groundwater or drainage water. The observed SRP flux at the outlet is within the uncertainty of the Radon analysis and the variability of groundwater concentrations. Therefore we do not expect considerable other sources than groundwater during the observed conditions but surely need to acknowledge the uncertainties in our calculations. We added this argumentation to the result section 4.1. Note that in summer 2019 discharge was even lower than 2020 (which also suggests a clear dominance of GW to total discharge) but SRP concentrations at the outlet were different in 2019 and 2020. This suggests variation in groundwater SRP concentration of the same contributing area in time or variation in groundwater SRP concentration in different contributing groundwater source areas. The older SRP groundwater data (Figure7) also suggest that SRP GW concentrations are not constant and may vary in space and time. Moreover we agree that during summer low flow conditions also

other in-stream/hyporheic processes may potentially lower SRP stream concentration, especially in 2020, but SRP fluxes in the downstream reach, where no substantial exchange between groundwater and stream water was observed, did not significantly change. This is in line with findings of Bernot et al. (2008) who observed only low SRP uptake in small agricultural streams with comparable low SRP concentrations. We consider this in the revised discussion section.

**Specific comments**

**Comment:** Again, language throughout could be greatly improved. I've pointed out only some examples (not exhaustive) below in technical comments. This applies not only to grammar but also to conceptual and technical language. I also have several specific comments below around technical aspects of the study which I think need refinement; these edits should also greatly aid the story of the paper.
A key point throughout this study is the connection of (deeper) groundwater to the stream under low flow conditions, supplying virtually all the flow, which coincides with generally the highest SRP concentrations. While I agree on this water source being prevalent at low flows, I'm not sure this study establishes this as the sole source for any given low flow period, as is illustrated with the two September campaigns. We only see detailed data for September 2020, which seems demonstrative of the groundwater P hypothesis, but notably September 2019 had much greater stream SRP concentrations (up to ~60-70 ppb). Unless deeper groundwater also increased SRP by 20-30 ppb, which I wouldn't expect as usually groundwater P is less dynamic, this high stream SRP leaves us wondering what the story is for September 2019 since groundwater SRP is likely lower. (Is there groundwater SRP data for this campaign too? I'm not seeing it.)

**Response:** This is a good point but again groundwater SRP concentrations are not constant and may vary in space and time. This can be seen in Figure 7. The boxplot shows that 75% of the SRP groundwater concentrations ranged between 0.03 and 0.07 mgPL$^{-1}$. This range still explains the measured data at the outlet for both campaigns. Note also the log-log-linear relationship between concentration and discharge in the three measurement campaigns shown in Fig. 7a. SRP concentration clearly show an increase with decreasing discharge. Assuming that the groundwater fraction in total stream discharge increases with decreasing discharge this appears plausible for us. Again: We do not explicitly exclude other sources of SRP but argue that groundwater is the dominant one. We did not measure SRP in the groundwater wells in 2019.

**Comment:** Further on the groundwater SRP: these groundwater SRP values are really high in general. I've read through the Wriedt et al. 2019 reference (mein Deutsch ist schlecht aber ich weiß genug) and I'd assume that the Schäfertal catchment would classify as part of their "Bergregion" (being part of the Harz mountains/Berge) – the Bergregion generally sees groundwater "phosphate" [note they analysed P via molybdenum-blue, i.e., it's SRP] concentrations of roughly 50 ppb as PO4 or less (hence, around 16 ppb or less as P) depending on depth, etc.. The Schäfertal catchment in this current study, however, seems to have much greater groundwater SRP concentrations than this natural reference of 16 ppb SRP (despite its oxidated status), with Figure 7 suggesting a median around 50 ppb P (more than 5- fold greater than the natural reference). Having worked in catchments with comparable geologies (greywacke), I would be floored by 50 ppb SRP being "natural" or "geogenic". This is all to say: I sincerely doubt the dominant source of P in the study stream as being "geogenic". I'm not suggesting that the text explores this point – as it's not the point of the study – but rather that the authors reckon with this and remove unnecessary/unsupported text from the Discussion, then refocus the text on how the proximal sources of P coupled with hydrological flowpaths determines in-stream SRP for their three sampling campaigns.

**Response:** Comparing the comments of the reviewer with our own statements in our discussion in section 3.5 and the conclusions on the contribution of agriculture and geogenic sources to groundwater SRP we think these viewpoints are not very different - we do not rule out agricultural

impact on groundwater SRP concentration. But we agree to change the wording in the abstract which suggested that geogenic sources are the overwhelming contributor of SRP to groundwater (see revised manuscript).

**Comment:** Relatedly, the authors argue for the catchment soils being highly P sorptive and thus not conducive to P leaching to groundwater (L467-468). A change in soil organic matter content and total P with depth is not necessarily an argument for limited soil P leaching (L467) since P sorption capacity is not a function of total P and is only somewhat related to organic matter. We'd need more direct observations of P sorption capacity, such as the degree of P saturation (DPS). In fact, the Kistner et al. (2013) in-text reference gives topsoil (0-5 cm) DPS in the Schäfertal of roughly 31% -- this is fairly high and saturated to the point that the soil is conducive to P loss in either surface or subsurface runoff (Fischer et al. 2017; Kleinman 2017). Further P loss in the subsurface soil would depend more on the soil's sorption chemistry, which is unavailable. So I am not convinced that soil P (particularly that coming from the historical fertiliser use in this predominantly agricultural catchment) here isn't reaching groundwater.

**Response:** There still seems to be some misunderstanding. What we wanted to point out is that the soils in the study watershed are highly sorptive and there are several indications which support this arguing, for example, if soils had limited sorption capacities, we would not expect such low SRP concentrations in drain water. This does not mean that no SRP from the soil column can reach the groundwater, and we did not state this in the discussion. The mean DPS value of roughly 31% was calculated according to van der Zee and Van Riemsdijk (1988). These values are in general lower than those calculated in Fischer et al. (2017) and Pöthig et al. (2010) and not directly comparable. Although WSP of around 13 mg P/kg indicate well P supplied soils for plant growth risk of leaching into groundwater was estimated low because of large distance between arable soils and groundwater head (Kistner 2007). Moreover half a year after P fertilizer application water soluble P content of the top soil was only 0.3% higher than those of unfertilized fields (Kistner 2007). This indicates high P sorption capacity of the soil which leads to a quick adsorption of readily available P compounds from the applied P fertilizer. Here we only argue that P leaching is likely low compared to other soil and aquifer properties. We still think that discussing these aspects is valuable for understanding SRP transport in such loamy lower mountain range catchments.

**Comment:** While it's ok (in my humble opinion) to use the term "soluble reactive P", it does seem silly to use that term but then call the unreactive P in the same filtered water sample as "dissolved organic P" (I'm assuming that's what "DOP" stands for, as it wasn't defined). Additionally, I'd be cautious about calling the unreactive P in a filtered sample "organic" – it likely is predominantly organic P but won't necessarily be 100% organic P (simply "unreactive", as in "molybdate-unreactive", is more accurate).

**Response:** While it is common using the molybdate-reactive P (soluble reactive P), it would be quite unusual referring to the remainig P fraction as "unreactive". We included into the method section: "We operationally define the difference of TDP and SRP as dissolved organic phosphorus (DOP), although this fraction may contain some inorganic phosphorus."

**Comment:** L234: Dissolved P (DP) was not defined in the methods (L266). Is this total dissolved P? If so, I suggest including "total" in the name (i.e. TDP).

**Response:** We now defined dissolved P (DP) as total dissolved P (TDP).

**Comment:** A focus of the paper is on baseflow conditions and the winter (January 2019) campaign is referred to as baseflow (L312). I think this needs to be more careful, as I'm not sure this event is really 'baseflow'. Discharge during the January campaign looks to me like a receding limb following a rain on snow event (Fig 2) – not exactly normal baseflow. Groundwater contribution as % of total

discharge would be somewhat lower then (groundwater meaning what your groundwater endmember represents). Instead, much more Q is likely coming from runoff and shallower stores (soil matrix). [In contrast, the summer September campaign is thoroughly deep groundwater, which the model results also support. This makes sense to me hydrologically and strikes me more as 'baseflow' as I understand it.] Perhaps the authors just need to clarify what is meant by 'baseflow'.

**Response:** We define baseflow as stream discharge, which is not containing direct runoff from a previous runoff event. It is the portion of the streamflow that is not directly generated from excess rainfall during a rain event, i.e., the streamflow that would exist without the contribution of direct runoff from the rainfall. This is a hydrological definition. Discharge in January campaign was relatively high and the reviewer is possibly right in assuming that some interflow, i.e., flow coming from shallower stores (e.g., soil matrix) from soils is also contributing to the discharge.

**Comment:** L301-302: So, the 'groundwater' endmember includes not only groundwater wells but also agricultural drains, and apparently these water sources have fairly similar Rn concentrations. This means it's difficult, based on the Rn data, to differentiate between these water sources to the stream. Is it possible at some points that the drains are delivering P to the stream? At very low baseflow, couldn't this P be remobilised into the water column? At the very least, it's confusing how the different subsurface sources of water sometimes get lumped into "groundwater".

**Response:** Generally, radon concentrations in water decrease (1) by decay and (2) by degassing. In an aquifer (i.e., in groundwater) decay is balanced by radon production and degassing does not occur. In the agricultural drains decay is NOT balanced by radon production but the residence time of the water within the drains is too short to allow notable decay. Degassing does not occur in the drains because the radon concentration in the air-filled headspace of the drain is in partition equilibrium with the drain water, or in some cases water-filled. Hence groundwater and drain water show the same radon concentration, which makes it reasonable that these two subsurface sources of water "get lumped into" a radon groundwater end-member. At the same time the reviewer is right in saying that with the radon method it is not possible to distinguish groundwater from the water produced by the drains, as they are essentially the same water, despite perhaps coming from different areas of the aquifer. We now use "lumped radon groundwater and tile drain water endmember" instead of "groundwater endmember" in connection with the Rn based calculation of subsurface inflows when a contribution of tile drain water cannot be excluded (winter campaign).

**Comment:** L308-312: This section should be deleted. First, it doesn't make sense to compare the Rn concentrations across different conditions anyways. The Rn tracer will degas back to atmosphere very differently across conditions. The summer low flow likely allowed tons of Rn degassing despite how much the GW contributes to Q. This Rn concentration comparison just has no place here. Second, the text here seems to imply that discharge in January was mostly GW while in September it was less so – the rest of the text argues the opposite. Absolute GW discharge could've been higher in January (although the data/model results suggest the GW Q's were pretty similar, which seems about right to me), but it made up proportionally *less* of the total Q.

**Response:** FIRST: We don't see the reviewers point why "it doesn't make sense to compare the Rn concentrations across different conditions". The differing degassing rates are explicitly accounted for by the FINIFLUX model. We don't simplify degassing by assuming a loss of "tons of Rn" but quantified the loss based on a specific degassing coefficient considered in FINIFLUX. The sensitivity of the modelled groundwater discharge rates on this applied degassing coefficient was assessed in detail (as discussed in the mns.). Hence, from our perspective the Rn concentration comparison makes perfect sense here.
SECOND: The message, which is clearly conveyed throughout the section is that river discharge in summer (September) was mostly GW while in winter (January) "normal baseflow" contributed about 60 – 65% of GW to the river discharge. This is the case because, on the one hand, GW discharge rates

were comparable (0.6 l/s vs. 0.75 l/s) but, on the other hand, river discharge rates differed and were considerably lower during summer. This shows that while in winter multiply flow paths and water sources contribute to total stream discharge, during draught conditions groundwater almost exclusively makes up stream flow.

**Comment:** L318-327: The sensitivity analysis here is nice to have. The text here demonstrates why I previously recommended giving an a priori value for the degassing coefficient based on the Raymond et al. 2012 review (see their empirical equations). I think this would bolster the discussion on this point. This is important because, for example, the degassing may be much greater than that parameterized for the winter event which implies that there must be greater 'groundwater' input (including the agricultural drains) in the first ~450 m which is also where the greatest increase in SRP flux occurs. I suspect the degassing may be too low because of the poor model fit in the downstream sections. Additionally, could a sensitivity analysis be performed for the summer campaign too?

**Response:** Comparing the various degassing equations is a good approach to better understand the uncertainties associated with the degassing coefficient (k) when calculating radon mass-balances, especially for small streams. The approach was executed (and discussed in detail) in the original study by Schubert et al., (2020) applying the commonly used equations by O'Connor and Dobbins (1958) and by Negulescu and Rojanski (1968). These two equations are often used in radon studies to capture the possible ranges in radon gas exchange (see e.g., Unland et al., 2013 or Atkinson et al., 2013). Unfortunately, neither of these equations captured the down-stream radon decrease, which is why we also attempted a propane injection and a fitting method, assuming downstream was either neutral or loosing as suggested by the water balance (see e.g., Cartwright et al., 2014). In the current paper we want to abstain from repeating the facts published in Schubert et al., (2020) but have made the best estimate of k from the different methods discussed there. Then conducted a sensitivity analysis for both summer and winter campaigns based on 25 % of this value, which seems reasonable since the empirical equations were not able to capture the radon loss in this very small stream (most degassing equations are made for larger rivers where sewage input and associated oxygen reduction was a problem historically). We added related text to the manuscript (see manuscript with track changes) . Still, generally the reviewer is right in saying that the higher the degassing, the more groundwater inflow is needed to compensate the mass-loss to the atmosphere.

**Comment:** L342: It's not necessarily that there was "intense radon degassing" during this low flow – degassing would be much more intense due to turbulence (a key influence of radon loss) during higher flow, no? I think it's just that stream velocity was very slow so the time available for degassing was much greater.

**Response:** Indeed, the long time period available for degassing increased radon loss by degassing between two measurement sites. However, the low flow conditions also lead to a very shallow water level in the creek (only cm).  And a very shallow water level is favorable for degassing because it results in a very high ratio water surface / water volume. The water column depth is a very important parameter, as seen by its inclusion in almost all empirical degassing equations. Generally, the depth is likely a significant uncertainty in the calculation of the degassing in particular for very shallow water levels, as it is difficult to measure and extrapolate it over the whole stream length.

**Comment:** Section 3.4: SRP:DOP means little, has no biogeochemical basis, and doesn't suit as a tracer; just focus on the SRP and DOP masses themselves. You can make the same point but with much more clarity by just focusing on DOP concentration – the ratio is just a distraction.

**Response:**  SRP and DOP (as defined) are both fractions of phosphorus. They differ (biogeochemically) in their origin (SRP can be released by zooplankton grazers, DOP can be released by anaerobic organic matter decomposition). The ratio of SRP/DOP is specific for different sources of P. The ratio can therefore be used to identify the source of phosphorus in the stream by comparing

the SRP/DOP ratio in the stream with the ratios in potential sources. A ratio tells much more about possible sources than concentrations of either SRP or DP, as concentrations can change rapidly by e.g. dilution, while ratios remain stable. The same principle is applied with other elements, e.g. the ratio of chloride to sulfate to characterize different groundwater bodies. We therefore think using the ratio of SRP to DOP is more convincing than concentrations of either SRP or DOP.

**Comment:** Section 3.4: This section repeatedly refers to porewaters from 7 cm deep as "shallow". Considering that stream itself is only 1 to 3 cm deep and that the substrate is fairly fine (L130), 7 cm below the substrate is actually very deep in the hyporheic zone, if it can be considered hyporheic at all (and remember: sediment porewater's connection to the water column is mostly via hyporheic exchange).
Contribution to total hyporheic exchange tails off with depth enormously so it's likely the waters down here move extremely slowly and represent a tiny minority of total hyporheic exchange flow. Boano et al. 2014 discusses this in detail. Any discussion of sediment should make it clear this is relatively *deep* sediment, and so sentences like L390-391 should be given better context. [Even with the reductive dissolution of Fe oxide bound P, it isn't clear what effect this has on the water column because this porewater must travel upwards through the hyporheic zone and likely comes across the zone of DO penetration and therefore potential for re-sorption.]

**Response:** We suggest to keep the description of the porewater from 7 cm deep as "shallow" because thickness of stream sediments was more than 20 cm although it is difficult to assess whether also deeper parts of the sediment are contributing to exchange with the stream water. Moreover one should keep in mind that water depth is highly variable in time and not limited to the range of 1-3 cm. We added this arguing also in the manuscript.

**Technical comments**

[Again, these aren't exhaustive. Please proofread carefully throughout.]
L23: "…often with the highest concentrations"
done
L27: "of SRP fluxes"
done
L28: revise to "… and sampled for SRP, iron, and 14C-DOC…"
done
L33: "and was thus…" – this logical connection doesn't make sense here because the abstract doesn't establish for the reader key points such as (1) SRP is high in groundwater and (2) summer low flow is dominated by GW (deeper GW, rather than other shallower sources). Please improve the connection.
done

L35-36: Remove "Examination of"; replace "confirms" with a verb more along the lines of "corroborates" or similar – temper the tone with scientific caution.
done
L37: I'm not sure what's being argued here with the bit on 'seepage from agricultural phosphorous'. [Note that P doesn't end with "ous"!] What does this mean to the reader at this point in the abstract?
done
L53: Stream sediments don't just provide P via reductive dissolution, there's also desorption (among other processes).
done
L56: I would advise against strongly scientific certain language like "undoubtedly". Could just revise this sentence to say "Subsurface transport is often dominated by preferential flow…"
done
L58: "therefore"

done

L59: How does a factor "such as soil P sorption saturation (Psat)" "greatly increase P mobility through soils" ? Do you mean, soils with a high P saturation?

Yes, you are right. We corrected accordingly

L69: Nitrate is an anion: NO3-

done

L74: I don't think "monomeric" is the term you mean here (I don't think phosphate counts as a monomer…). Do you mean "orthophosphate", i.e., phosphate in its various hydrated states (HPO4 2-, H2PO4 1-, etc.)?

**Response:** Because the notion monomeric is possibly not clear, we took it out.

**Comment:** L75: "Molybdate reactive P redox-mediated release" – this isn't clear as written; please proofread throughout. Also, I pointed out in the previous review that it does not make sense to use 'molybdate reactive P' in one section but 'soluble reactive P' or other variants elsewhere. The 'reactive' implies 'molybdate reactive'. It's fine to me to use 'molybdate reactive P' in the manuscript but if so, use it throughout to avoid confusion. (The literature is full of varying definitions of the same analyte; let's not muddy the waters more.)

**Response:** We agree with the reviewer and use SRP consistently within the whole manuscript.

L93: Rather than 'level', just use 'concentration' for consistency.

done

L129: 1747 meters?

This is correct

L130: Delete 'fraction'

done

L131: Revise the forest area part for grammar.

done

L132: TP is not defined.

done

L136: "DSP" – was this supposed to be 'DPS'?

done

**Comment:** L158: "runoff" seems like the wrong term here; consider "…under these drought conditions, water from deeper, older storage contributed more to stream discharge, particularly after a wet season [or in comparison to a wet season?]."

**Response:** runoff is also often used for the sum of all components of water entering the stream but sometimes also restricted to water from the surface of an area of land, therefore we now use the term water. Furthermore "particularly after a wet season" is correct.

**Comment:** L163: Edit to: "Baseflow data show clear seasonal variations, with NO3 peaking in winter while DOC and SRP peak in summer." [is the Dupas et al. 2017 reference specific to your study stream? Does it contain this data? Unless the answer to both is 'yes', then remove the reference]

**Response:** done; yes, Dupas et al. 2017b is referring to the study stream. Thanks for that because we missed to add the second manuscript of Dupas et al. from 2017 and we now distinguish between Dupas et al. 2017a and Dupas et al. 2017b.

**Comment:** Should the title of section 2.2.1 be broader than just "lateral inflows"? E.g. 2.2.1.1. doesn't really cover "lateral" inflows, but stream discharge itself. Maybe "Stream hydrological measurements" in general, or similar.

**Response:** good point because 2.2.1 is related to water quantity and 2.2.2 is related to water quality.

Our idea was to indicate the importance of lateral inflow and its spatial distribution along the stream but of course we also consider vertical inflows through the stream bottom. We select "Water inflows to the stream" to still consider the spatial variation of inflows to the stream.

**Comment:** L212-213: This sentence adds little here. Instead, could you add here or perhaps in the results what gas transfer velocity could be expected (independent of your model) for your study stream based on the relationships provided by Raymond et al. 2012? I.e. give a value or range of plausible values. This would provide confidence in the parameter value obtained via your FINIFLUX model calibration.

**Response:** Generally, all published empirical equations that aim at quantifying k (and thus, the ones presented by Raymond et al.) are derived by fitting degassing data from a wide range of different "model streams". However, none of these model streams are as small as the Schäfertal stream. That makes the Raymond equations only suitable to only a limited extent for parameterizing radon degassing in our case. However, as requested by the reviewer we added the resulting information to the manuscript. We made considerable changes in section 3.3 of the manuscript.

**Comment:** Section 2.2.1.2. This section needs to be more clearly organized and probably should feature some paragraph breaks. The point raised on L209 ("A crucial parameter… is the rate of radon degassing…") just gets brought up but is left unresolved.

**Response:** Section 2.2.1.2. (L209): We tried to give the section a clearer structure by adding some paragraph breaks. However, we don't agree that the rate of radon degassing "*just gets brought up but is left unresolved*" in the text. Radon degassing is sufficiently discussed in the section 3.3 of the manuscript or the related literature is cited.

L217: "traces" – "tracer"
done
L226: Item (iv) is unclear to me.
**Response:** Both item (iv) and item (v) relate to $^{222}$Rn activities; i.e., (iv) $^{222}$Rn in stream water specific for the sub-section and (v) $^{222}$Rn in the overall groundwater.

L235: "soluble reactive P (SRP)"
done
L239: switching between "Fe" and "iron"…
**Response:** we used now Fe throughout the text with the exception of "iron–reductive conditions"
L244: remove the comma after "both"
done
L253: italicize the "g" for the gravitational constant
done
Figure 5: The axis on the right side has a "0.0" label – this is inconsistent with the log scale. Edit this axis to be more like that in Figure 6B.
done

**Comment:** L339-340: This sentence confused me because I knew the January 2019 total discharge was much greater (5 to 6 L/s) until I realised this sentence referred to the water gained within the study reach – perhaps edit to make this clearer.
**Response:** We clarified this sentence in section 3.3.

L341: 'stream depth' instead of 'water level'
done

**Comment:** L342: It's not necessarily that there was "intense radon degassing" during this low flow – degassing would be much more intense due to turbulence (a key influence of radon loss) during higher flow, no? I think it's just that stream velocity was very slow so the time available for degassing was much greater.

**Response**: Indeed, the long time period available for degassing increased radon loss by degassing between two measurement sites. However, the low flow conditions also lead to a very shallow water level in the creek (only cm).  And a very shallow water level is favorable for degassing because it results in a very high ratio water surface / water volume. The water column depth is a very important parameter, as seen by its inclusion in almost all empirical degassing equations. Generally, the depth is likely a significant uncertainty in the calculation of the degassing in particular for very shallow water levels, as it is difficult to measure and extrapolate it over the whole stream length.

**Comment:** L356: Why isn't electric conductivity included in Table 1? This is actually the closest thing to a 'conservative tracer' and can help compare the different waters. Table 1: It's not clear what the distinction is between the two groundwater entries are (perhaps explain in caption?). Also, maybe it's just the formatting, but should the second GW entry have "n=6" as the station?

**Response:** Electric conductivity was measured in surface water only but not in groundwater and in sediment pore water. Thus we do not report it in Table 1 but we give the values which were not very different in upstream and downstream station in the text. n=6 was added to the table.

**Comment:** L388: Before this point (e.g. L232), I was already wondering why nitrate wasn't in Table 1 but sentences like this especially highlight this – can nitrate be added?
**Response:** Yes, we added the nitrate values but without groundwater wells

**Comment:** Figure 7: Clarify the sources for the historical data here (they're not from this study, correct?) in the caption.
**Response:** we added the information on the data to the Figure 7.

**Additional references**
Boano, Fulvio, Judson W Harvey, A. Marion, Aaron I Packman, R. Revelli, L. Ridolfi, and A. Wörman. "Hyporheic Flow and Transport Processes: Mechanisms, Models, and Biogeochemical Implications." Reviews of Geophysics 52 (2014): 603–79. https://doi.org/10.1002/2012RG000417.Received.

Fischer, P., R. Pöthig, and M. Venohr. "The Degree of Phosphorus Saturation of Agricultural Soils in Germany: Current and Future Risk of Diffuse P Loss and Implications for Soil P Management in Europe." Science of The Total Environment 599–600 (December 1, 2017): 1130–39. https://doi.org/10.1016/j.scitotenv.2017.03.143.

Kleinman, Peter J. A. "The Persistent Environmental Relevance of Soil Phosphorus Sorption Saturation." Current Pollution Reports 3, no. 2 (June 1, 2017): 141–50. https://doi.org/10.1007/s40726-017-0058-4.

**Reviewer 2**

**Comment:** I have looked through the responses to Reviewers 1 and 2 and the authors have done a thorough job in addressing the previous two reviewers' comments and concerns.

This manuscript addresses an important subject: understanding and apportioning P sources under baseflow periods of greatest ecological sensitivity. Many papers have assumed that elevated P concentrations under baseflow reflect point source contributions. By combining salt dilution testing and novel 222Rn and 14C-DOC measurements, with more routine water quality analyses, the authors have been able to demonstrate that groundwater is a dominant source of SRP to an upland headwater stream. I enjoyed reading the manuscript and I recommend the paper for publication, subject to some small revisions:

**Response:** many thanks for this general positive comment

**Comment:** Line 180: "gross gains, gross losses and net change"– please specify "in flow" or "water flux"

**Response**: This refers to water flow and was added to this sentence.

**Comment:** Line 234 – It would be helpful if the authors could specify what they measure as DP. I assume that this is, in fact, a total dissolved P (TDP) fraction, using the same digestion step as for TP, but for a filtered sample? It would be helpful to at least provide some very concise information here about analytical methods, so that the reader can clearly differentiate these two dissolved/soluble P fractions.

**Response**: We now defined dissolved P (DP) as total dissolved P (TDP).

Line 355 – Please change "steam" to "stream"; also conductivity units should be mS cm-1?

done

Line 424 – "PO4+" should be "PO43-"

done

Line 439 – I suggest changing this to "driven by within-river biogeochemical processes"

**Response:** we choose "in-stream biogeochemical processes" because the term "river" would be misleading for the small creek.

**Comment:** Line 444-445 – As far as I am aware, the Jarvie et al 2008 paper deals with one UK rural catchment where elevated baseflow P concentrations were attributed to agricultural point sources and septic tank sources, not groundwater. The statement, as currently written, incorrectly implies that groundwater is a dominant source of P in UK rural catchments. This is misleading; however, in *some* locations groundwater can be a source of ecologically-significant P concentrations. The authors may find the following reference useful: Holman et al (2008): "Phosphorus in Groundwater— An Overlooked Contributor to Eutrophication?" Hydrological Processes 22(26): 5121 – 5127.

**Response:** we agree with the reviewer and modified this sentence accordingly to prevent misunderstanding and also added the valuable reference of Holman et al. 2008.

Line 455 Please change "Nitrates" to "Nitrate"

done

Lines 500-503: "Currently enriched P in agricultural soils…..McCrackin et al 201)". This the first time in the manuscript that these issues about agricultural soils as long-term legacy P sources are raised. This is not a conclusion of this study and I recommend that this section of the Conclusions be moved to the Discussion.

done

Minor comment: throughout the manuscript, subscripts and superscripts were often omitted (mg L-1, NO3, NH4, etc.)

done

---

## Author Response (AR3)

**Revision notes of the manuscript**

Kindly note that in the revision notes, comments from the reviewers are marked with "**Comment**", while our responses are marked with "**Response**".
* * *
**Responses to the comments from reviewers:**

**General comments**

I commend the authors for their revision of this manuscript. The study is valuable for the literature on the interaction between hydrology and P biogeochemistry, and deserves to be communicated as effectively as possible.

The manuscript is much clearer throughout. Here are a few remaining points to be ultra-clear about:

**Comment:**

In the discussion on Rn degassing, its impacts on GW measurements, etc. – shouldn't the estimate of k in this study be reported? Pointing out that prior work (e.g. Raymond et al. 2012) uses much bigger streams is fair, but hopefully the ranges for k on L334-335 are still plausible. What k did FINIFLUX estimate? Note also that L234 calls it k_Rn but k is used elsewhere.

**Response:**

In the revised manuscript we describe the challenges related to the quantification of radon degassing (i.e., the k value) based on empirical equations in more detail. The k values from our FINIFLUX model calculations are now reported in an additional paragraph. As described in detail in Frei and Gilfedder (2015) and Schubert et al. (2020) (both cited in the "methods" section of the paper). FINIFLUX is using the empirical equation from O'Connor and Dobbins (1958) (O'Connor, D.J., Dobbins, W.E.: Mechanism of reaeration in natural streams, Transactions of the American Society of Civil Engineers, 123, 641-666, 1958.). As suggested by the reviewer, the k values resulting from FINIFLUX are compared to k values resulting from alternative empirical equations (Raymond et al., 2012). The differences in the results are discussed.

"k_Rn" in L234 was changed to "k".

**Comment:**

The new text added around the summer radon / GW modeling results (L344-358) certainly raises questions about the FINIFLUX results (primarily, how the sensitivity analyses give lower discharges than what was calibrated). However the authors discuss the challenges involved. (This could perhaps serve as a good case study for future development of FINIFLUX.) Just some food for thought: could uncertainties in the stream discharge measurements (via salt injections) also be involved? Is it possible that some sections were losing water to GW?

**Response:**

We did not investigate uncertainties in discharge associated with salt injections but if these uncertainties would be available they could in principle be considered using the FINIFLUX

model. As shown in Fig. 4 especially the January campaign shows clear losing conditions in the most downstream section.

**Comment:**
The GW P fluxes compared to stream P flux (L412-414) is a nice addition to the text. I would caution that the relatively wide range in uncertainty for the GW P fluxes leaves room for interpretation. Assuming the true GW P flux is contained within that bracket, what if the summer GW P flux was 4 mg s-1? Or 64 mg s-1? (Roughly 4-fold less and 4-fold more, respectively, of the summer stream P flux.)

**Response:**
The range of calculated GW-P flux is based on the variability of SRP concentrations in individual groundwater wells. This range represents the maximum range if we assume that the GW P flux can be calculated using the lowest or highest concentration in a well. Because the GW P flux is most likely a mixture of different GW SRP concentrations, GW P fluxes calculated with concentrations near the mean GW SRP concentrations are much more likely. Therefore, in our view, the reported maximum ranges of GW P fluxes represent rather theoretical maximum ranges. The P fluxes at the catchment outlet agree quite well with those based on mean GW SRP concentrations.

**Comment:**
The Discussion could use some revision for efficiency and clarity. Some examples: the sentence beginning on L423 reads like a repeat of L417-418. The first half of section 4.2 (L451-488) could be condensed/simplified. I don't follow how the "findings confirm… that SRP concentration in the saturated zone is controlled by sorption equilibrium under oxidizing conditions in the upper groundwater" (L517-518).

**Response:**
We agree with the comment and shortened the discussion accordingly and took the sentence from L423 out. For setting our results in a broader context of earlier investigations we think that it is necessary to present those findings in the suggested detail and we therefore would like to keep this discussion mostly as it is. Nevertheless we shortened this sections slightly (L486).
We rewrote the sentence (L517-518) as follows: Our findings suggest that SRP concentrations in the saturated zone is controlled by oxidizing conditions in the upper groundwater, and that SRP losses through seepage are largely buffered.

**Comment:**
L538 suggests sediment P fluxes were quantified; I don't believe they were, but rather suggested to likely be low. Please clarify.

**Response:**
We rewrote the sentence as follows:
Because of the limited quantitative P exchange between stream sediments and stream water, stream sediments that may have originated from agricultural soils eroded into the stream did not contribute significantly to the SRP loads exported from the stream.

**Comment:**

I suggest reading through the manuscript once more with fresh eyes for small fixes. Below are a few things I noticed.

DOC needs to be defined in the abstract.
Suggest defining P on L42.
L208: 'was' instead of 'were'
L216: "…stream is provided in"
L336: 'ran' instead 'run'
L380: DOP
L421: "…SRP-concentrations, TDP, …" – odd wording
L438: remove the "that"
L505: van Dael, not van Deal
L515: Clarify the 0.05 mg/L reference (mean/median?)
Conclusions should now be Section 5
Some references are repeated.

**Response:**

All done with the exception of L42 and the manuscript has been checked again.